

# Changes in gross oxygen production, net oxygen production, and air-water gas exchange during seasonal ice melt in the Bras d'Or Lake, a Canadian estuary

Cara C. Manning[1,2,3,a], Rachel H. R. Stanley[4], David P. Nicholson[2], Brice Loose[5], Ann Lovely[5], Peter Schlosser[6,7,8], and Bruce G. Hatcher[9]

[1]MIT/WHOI Joint Program in Oceanography/Applied Ocean Science and Engineering, Woods Hole, MA, USA.
[2]Department of Marine Chemistry and Geochemistry, Woods Hole Oceanographic Institution, Woods Hole, MA, USA.
[3]Department of Earth, Atmospheric and Planetary Sciences, Massachusetts Institute of Technology, Cambridge, MA, USA.
[4]Department of Chemistry, Wellesley College, Wellesley, MA, USA.
[5]Graduate School of Oceanography, University of Rhode Island, Naragansett, RI, USA.
[6]Lamont-Doherty Earth Observatory of Columbia University, Palisades, NY, USA
[7]Department of Earth and Environmental Sciences, Columbia University, Palisades, NY, USA
[8]Department of Earth and Environmental Engineering, Columbia University, New York, NY, USA
[9]Department of Biology and Bras d'Or Institute, Cape Breton University, Sydney, NS, Canada.
[a]Now at: Department of Earth, Ocean and Atmospheric Sciences, University of British Columbia, Vancouver, BC, Canada.

*Correspondence to:* Cara C. Manning (cmanning@eoas.ubc.ca)

**Abstract.** Sea ice is an important control on gas exchange and primary production in polar regions. We measured net oxygen production and gross oxygen production using near-continuous measurements of the $O_2/Ar$ gas ratio and discrete measurements of the triple isotopic composition of $O_2$ in the Bras d'Or Lake, an estuary in Nova Scotia, Canada, as the bay transitioned from ice-covered to ice-free conditions. The volumetric gross oxygen production was $5.4(^{+2.8}_{-1.6})$ mmol $O_2$ m$^{-3}$ d$^{-1}$, similar at the

beginning and end of the time-series, and likely peaked at the end of the ice melt period. Net oxygen production displayed more temporal variability and the system was on average net autotrophic during ice melt and net heterotrophic following the ice melt. We performed the first field-based dual tracer release experiment in ice-covered water to quantify air-water gas exchange. The gas transfer velocity at >90 % ice cover was 6 % of the rate for nearly ice-free conditions. Published studies have shown a wide range of results for gas transfer velocity in the presence of ice, and this study indicates that gas transfer through ice is much

slower than the rate of gas transfer through open water. The results also indicate that both primary producers and heterotrophs are active in Whycocomagh Bay during spring while it is covered in ice.

## 1   Introduction

The annual cycle of sea ice formation and melt regulates primary production and $CO_2$ uptake and ventilation in polar regions. Ice alters the rate of air-water gas exchange, reduces the penetration of light into surface water, changes stratification and mixing

processes, and harbors microbes and biogenic gases including $CO_2$ (Cota, 1985; Loose et al., 2011a; Loose and Schlosser, 2011).





The question of whether climate change will increase or decrease Arctic Ocean carbon uptake is a topic of considerable debate (Bates et al., 2006; Bates and Mathis, 2009; Cai, 2011). Global warming is causing dramatic reductions in sea ice cover and increases in freshwater inflow and organic carbon supply to the Arctic Ocean, which impacts ecosystems (ACIA, 2004; Vaughan et al., 2013; Macdonald et al., 2015). Because conducting field work in the Arctic is challenging, measurements of

productivity and gas exchange are limited. Biogeochemical time-series observations resolving seasonal changes in productivity are particularly scarce in the Arctic (MacGilchrist et al., 2014; Stanley et al., 2015). Measurements at Palmer Station in Antarctica show a strong seasonality in biological productivity and carbon uptake associated with changes in light, physical mixing, and grazing and demonstrate the benefits of high-frequency sampling for quantifying $CO_2$ uptake in seasonally ice-covered waters(Ducklow et al., 2013; Tortell et al., 2014; Goldman et al., 2015).

Parameterization of gas exchange in the presence of ice also remains highly uncertain. Many investigators have assumed that there is negligible gas transfer through ice and therefore the gas transfer velocity can be linearly scaled as a function of the fraction of open water, multiplied by the open water gas transfer velocity (Takahashi et al., 2009; Legge et al., 2015; Evans et al., 2015; Stanley et al., 2015), however, this assumption is not well verified. A recent study by Butterworth and Miller (2016) concluded that gas transfer velocities in the Southern Ocean at 0–100 % ice cover approximated a linear relationship

with ice cover, but other studies report that gas exchange is reduced or enhanced in the presence of sea ice relative to a linear scaling based on the fraction of open water, including some studies measuring higher transfer velocities in ice-covered waters than in open water (Fanning and Torres, 1991; Else et al., 2011; Papakyriakou and Miller, 2011; Rutgers van der Loeff et al., 2014). Additional studies show that gas exchange in ice-covered waters cannot be predicted from wind speed alone, which may be a cause of the wide range of results (Loose et al., 2009; Lovely et al., 2015; Loose et al., 2016).

In this study, we measured productivity and gas exchange over a 1-month period during and following ice melt in the Bras d'Or Lake, an estuary on Cape Breton Island in Nova Scotia, Canada. The Bras d'Or Lake consists of several interconnected channels and basins and has a surface area of 1070 $km^2$ and an average depth of ∼30 m (maximum 280 m) (Petrie and Raymond, 2002; Petrie and Bugden, 2002). The Bras d'Or Lake exchanges water with the Atlantic Ocean ocean primarily through the Great Bras d'Or Channel at the northeastern region of the estuary; this channel has a shallow (16 m deep) and narrow (0.3

km) restriction at the mouth (Petrie and Raymond, 2002). We conducted the work for this study in Whycocomagh Bay, an enclosed embayment approximately 13 km long and 3 km wide, at the western end of the estuary, approximately 60 km from the open ocean (Figure 1). Whycocomagh Bay is separated from the rest of the Bras d'Or Lake by Little Narrows, a channel which is approximately 0.2 km wide and 0.5 km long. Whycocomagh Bay is up to 40 m deep and Little Narrows channel is ∼15–20 m deep (Gurbutt and Petrie, 1995). Maximum ice cover is typically reached in early March and ice disappears most

rapidly in April and has usually disappeared completely by the first week of May (Petrie and Bugden, 2002).

We performed two dual tracer release experiments to quantify air-water gas exchange by adding $^3$He and $SF_6$ to the mixed layer. The first tracer release experiment (Injection 1) occurred under near-complete ice cover and the second experiment (Injection 2) occurred in nearly ice-free conditions. We measured net oxygen production (NOP) and gross oxygen production (GOP) at Little Narrows while Whycocomagh Bay transitioned from completely ice-covered to ice-free conditions. GOP

is the total amount of $O_2$ generated by autotrophic microbes as a result of photosynthesis. NOP is GOP minus respiratory



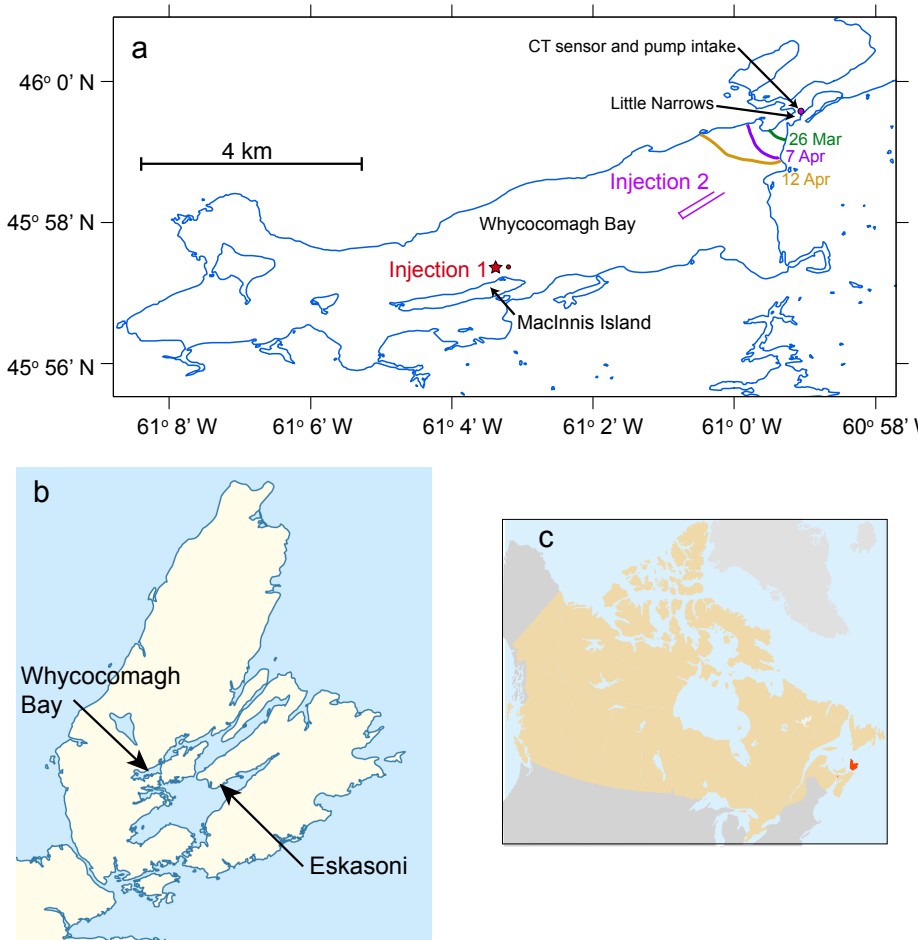

**Figure 1.** (a) Map of Whycocomagh Bay showing the locations of Injections 1 and 2 and the sampling equipment at Little Narrows. For Injection 1, the injection location is shown with a red star and the location where initial samples were collected is shown with a red circle. The colored lines labeled 26 Mar, 7 Apr, and 12 Apr show the location of the ice edge on these days. (b) Map of Cape Breton Island, showing the location of Whycocomagh Bay and Eskasoni (where wind speeds were obtained). (c) Map of Canada showing location of Cape Breton Island.





consumption of $O_2$ by autotrophs and heterotrophs. The ratio of these two terms provides an estimate of the export efficiency, i.e., the fraction of autotrophic production available for export from the mixed layer. The time-series approach allowed us to quantify non-steady state $O_2$ fluxes, which can be a significant fraction of the total $O_2$ flux in many regions (Hamme et al., 2012; Tortell et al., 2014; Manning et al., 2017b). We quantify GOP with discrete measurements of the triple oxygen isotopic
composition of $O_2$ (Juranek and Quay, 2013) and NOP with near-continuous measurements of the $O_2/Ar$ saturation anomaly (Cassar et al., 2009).

To our knowledge, there are no other published field-based experiments where the dual tracer technique was used in the presence of ice, and this study adds to a limited number of in situ measurements of NOP and GOP during ice melt (Goldman et al., 2015; Stanley et al., 2015; Eveleth et al., 2016).

## 10   2   Field work and analytical methods

### 2.1   Setup at Little Narrows

We continuously sampled water at Little Narrows channel (Figure 1) over a 1-month period (25 March–28 April 2013). We moored a Goulds SB Bruiser 5-18 GPM submersible pump with intake at ~0.5 m depth, placed inside a mesh filter bag to prevent large particles from clogging the pump, and a conductivity and temperature (CT) sensor (Sea-Bird Electronics SBE37)
at ~0.5 m depth. From the submersible pump, water flowed through flexible high-pressure PVC tubing submerged underwater to a 3-port pressure-relief valve (on shore) and was then pumped along shore (~50 m) to our sampling apparatus. The water passed through three 10" canister filters (100, 20, and 5 $\mu$m nominal pore size) and then into a sampling manifold containing valves for distributing water for measurement of $O_2/Ar$ (continuously, by mass spectrometry) and for discrete sampling. As discussed below, we sampled discretely for $SF_6$, $^3He$, and $O_2/Ar$ and the triple oxygen isotopic composition of $O_2$, and near-
continuously for $O_2/Ar$. Excess water flowed through tubing back into the bay. We covered the tubing on shore and the filter canisters in foam insulation to minimize temperature changes in the water. We deployed a Nortek acoustic doppler current profiler (ADCP) at 4 m depth in the middle of the channel beginning on 7 April through the end of the time-series. The ADCP measurements are not used to interpret the results below because our measurements did not display any correlation with tidal cycles. However, a figure of the ADCP data is provided as Supplemental Figure 1.

The CT sensor was initially placed closer to shore than the water pump because the cable was not long enough to reach the pump, but after obtaining a longer cable, we were able to co-locate the CT sensor with the water pump (beginning 12 April). For the discrete samples, we used the CT sensor temperature and salinity measurements beginning on 12 April (when we moved the CT to the same location as the pump) to calculate the equilibrium concentration of each gas. Prior to 12 April, we collected measurements with a YSI temperature and salinity probe from the water on shore and used these measurements
as the temperature and salinity for the discrete samples. We determined the average warming through the underway line to be 0.37(0.22) °C based on comparisons between the temperatures from the CT sensor (in situ) and the YSI probe (on shore) after 12 April and applied this offset to all YSI temperature measurements. For the continuous $O_2/Ar$ data we used a temperature record from a thermocouple in the bucket because it had fewer gaps in time. We calibrated the thermocouple using an Aanderaa



4330 optode sensor which contains a temperature sensor (accuracy $\pm0.03$ °C) and then decreased the temperature by 0.37 °C to correct for warming.

We installed the gas chromatograph (for measurement of $SF_6$) and mass spectrometer (for measurement of $O_2/Ar$) inside a garage connected to the Little Narrows Ferry building. The majority of the wet equipment was set up outside the garage adjacent to a window on the garage, which was used for connecting equipment and power cables between the outdoors and indoors.

We deployed the water pump and in situ CT sensor to the east (oceanward) of the Little Narrows cable ferry which periodically crosses the channel and operates 24 hours per day. We found no correlations between our temperature, salinity, and geochemical measurements and the position of the ferry within the channel. We collected conductivity, temperature and depth (CTD) profiles with a SBE 19Plus sensor at Little Narrows, usually by boat using a winch, but occasionally by lowering the CTD by hand on a rope from the Little Narrows cable ferry. The GPS-equipped boat enabled us to map out the location of the ice edge nearest to Little Narrows, to perform the second tracer injection, and to sample after the tracer injection.

## 2.2 Tracer injections

The approach in this study was to dissolve the tracer mixture ($^3$He/$SF_6$) in Whycocomagh Bay, and sample continuously at Little Narrows, a constriction at the mouth of the bay. The net surface flow within Whycocomagh Bay, Little Narrows, and St. Patrick's Channel is toward the ocean due to the substantial freshwater inputs to the bay (Gurbutt and Petrie, 1995; Yang et al., 2007) and therefore tracer dissolved within the bay at the surface will eventually pass through Little Narrows, or be ventilated to the atmosphere. Two tracer injections occurred during the time-series, resulting in estimates of the gas transfer velocity for two extremes: near-complete ice cover, and essentially ice-free conditions.

Injection 1 occurred through a hole in the ice from 30–31 March, near MacInnis Island (Figure 1a). Approximately 0.11 mol $SF_6$ and 4.0 x $10^{-4}$ mol $^3$He was diluted by a factor of 50 with $N_2$ and then bubbled using a manifold within the mixed layer, over a 21-hour period. We sampled for initial $^3$He and $SF_6$ concentrations after terminating the tracer addition, by drilling a separate hole near the injection site. Subsequently, we sampled the tracer as it flowed through Little Narrows from 7–11 April. From 31 March–11 April, the bay was nearly completely full of ice, with a small opening near Little Narrows (Figure 1a).

Injection 2 occurred by boat on the morning of 19 April. By this time, the bay was nearly ice-free and all tracer from the previous experiment had passed through and/or been ventilated to the atmosphere such that the tracer concentrations at Little Narrows were below detection. While the boat was moving, we used the same manifold as for Injection 1 to bubble approximately approximately 4.4 mol $SF_6$ and 0.021 mol $^3$He, diluted by a factor of four with $N_2$ into the mixed layer (Figure 1a). The injection lasted 40 min. We detected the tracers at Little Narrows beginning on 20 April 23:30 and measured the change in the ratio between 20 April–23 April as the tracer patch flowed through Little Narrows.

## 2.3 Measurement of $O_2/Ar$ and the triple oxygen isotopic composition of $O_2$

We set up an equilibrator inlet mass spectrometer (EIMS) for measurement of $O_2/Ar$ similarly to the system described in Cassar et al. (2009). However, we used a larger membrane contactor cartridge (Liqui-Cel MiniModule 1.7 x 5.5) because the design is





more robust than the Liqui-Cel MicroModule 0.75 x 1 used by Cassar et al. (2009). The water flow rate through the cartridge was ~1.5 L min$^{-1}$. We attached a custom female Luer-Lok fitting paired to a capillary adapter to the upper headspace sampling port and the lower sampling port was left closed.

For $O_2$/Ar and the triple oxygen isotopic composition of $O_2$, we collected samples in pre-evacuated, pre-poisoned glass

flasks from a spigot in the water pumped to shore, or for shipboard sampling, using a small submersible water pump. Analysis occurred within ~6 months of flask evacuation and 4 months of sample collection at the Woods Hole Oceanographic Institution with a Thermo Fisher Scientific MAT 253 isotope ratio mass spectrometer, following the method of Barkan and Luz (2003) with modifications as described in Stanley et al. (2010, 2015).

The precision of the discrete samples, calculated based on the standard deviation of equilibrated water samples run daily

along with the environmental samples was 0.011 and 0.020 ‰ for $\delta^{17}O$ and $\delta^{18}O$ respectively, 5.6 per meg for $^{17}\Delta$, and 0.07 % for $O_2$/Ar. The mean difference between the EIMS and discrete samples was 0.05 %, and the mean magnitude of the difference was 0.35 %; given the small mean offset, the EIMS data was not calibrated using discrete samples.

## 2.4  Measurement of $SF_6$

For $SF_6$, we collected ~20 mL water samples in 50 mL glass gas-tight syringes, then added ~20 mL of nitrogen and allowed

the samples to be shaken for 10 min to achieve equilibration between the headspace and water (Wanninkhof et al., 1987). After the water equilibrated to room temperature, we injected 1 mL of the headspace into an SRI-8610C gas chromatograph with an electron capture detector, heated to 300 °C (Lovely et al., 2015). We calibrated the detector response using a 150 ppt $SF_6$ standard (balance $N_2$).

We also operated an automated gas extraction system at Little Narrows which sampled nearly every hour. The system

recirculated a 118 mL loop of water through a membrane contactor, and the headspace of the membrane contactor was under vacuum, causing the gas to be extracted from the water into the headspace. This system enabled determination of when the tracer was flowing through Little Narrows and helped inform the timing of discrete sample collection.

$SF_6$ equilibrium solubility concentrations are calculated following Bullister et al. (2002) and diffusivity is from King and Saltzman (1995). Precision of the system, based on the standard deviation of measurements of the 150 ppt standard, was 7 %.

We assume a dry atmospheric mole fraction of 8 ppt for $SF_6$ (Bullister, 2015).

## 2.5  Measurement of $^3$He

For $^3$He analysis, we collected samples in copper tubes, mounted in aluminum channels and sealed the samples at each end using clamps. Sample analysis occurred at the Lamont Doherty Earth Observatory, using a VG5400 mass spectrometer for $^3$He and $^4$He concentration (Ludin et al., 1998). Error for $^3$He sample analysis (combined precision and accuracy) was ≤2 % of the

measured $^3$He excess concentration above equilibrium. We used He solubility from Lott and Jenkins (personal communication, 2015) and diffusivity from Jähne et al. (1987a). The Lott and Jenkins is solubility ~2 % higher than published data from Weiss (1971). The He solubility is for bulk He and we calculate the $^3$He solubility using an atmosperic mole ratio $M(^3He)/M(^4He) = 1.399x10^{-6}$ (Mamyrin et al., 1970; Porcelli et al., 2002), although some more recent results indicate the current ratio may be





slightly lower, $\simeq 1.390 \times 10^{-6}$ (Brennwald et al., 2013). We use the He equilibrium isotopic fractionation as described in Benson and Krause (1980b).

## 3 Calculations, results, and discussion

The three goals of the experiment were to 1) quantify gas transfer velocity by dual tracer release, 2) quantify gross oxygen production from the triple isotopic composition of $O_2$, and 3) quantify net oxygen production from $O_2/Ar$. We begin by discussing the hydrographic characteristics of the study area and then describe the calculations, results, and interpretation for each of the three goals, in sequence.

### 3.1 Hydrography

We began sampling $O_2$ at Little Narrows on 25 March, when Whycocomagh Bay was nearly (>95 %) full of ice, and completed sampling on 28 April, when the bay was completely ice-free (Figure 2). The surface ice cover retreated most rapidly between 16–18 April and was completely gone by 22 April or perhaps even earlier. Figure 2 shows 18 April and 23 April; MODIS images on 22 April were also ice-free (but more blurry, so are not shown in the figure) and we estimated ice cover to be <10 % in the bay during surveys by boat during daytime on 20 April. The ice was likely melting even at the beginning of the time-series since the surface water temperature was always above the freezing temperature of water (Figure 3). Changes in surface ice cover and total ice volume are both important factors during the study; changes in ice volume/thickness will affect stratification and convection in the mixed layer as well as light penetration through the ice, and the surface ice cover affects the rate of gas exchange (Smith and Morison, 1993; Butterworth and Miller, 2016; Loose et al., 2016)

CTD profiles at Little Narrows channel near the water pump intake showed substantial changes in stratification during the time-series (Figure 3). From 25 March through 8 April the water column was strongly stratified and we estimated the mixed layer depth to average 0.8(0.3) m. During this period, it was often difficult to determine the exact mixed layer depth because the mixed layer depth was similar to the length of the CTD and obtaining a stable CTD response so close to the surface was challenging. Following 8 April, the mixed layer deepened and warmed and its salinity increased, likely due to convection and heating following sea ice melt. For this period, we defined the mixed layer depth as the first depth where the density is 1 kg $m^{-3}$ greater than the value at 1 m. The mixed layer reached a maximum of 3.0 m on 23 April and then shoaled by the end of the time-series on 28 April. On most days, the density profile and mixed layer depth were driven by stratification in salinity, but for the final profile on 28 April the mixed layer depth was determined by a combination of temperature and salinity stratification due to heat uptake by the surface water following the ice melt. These changes in mixed layer depth must be considered in order to interpret the $O_2$ measurements and to quantify the gas transfer velocity.



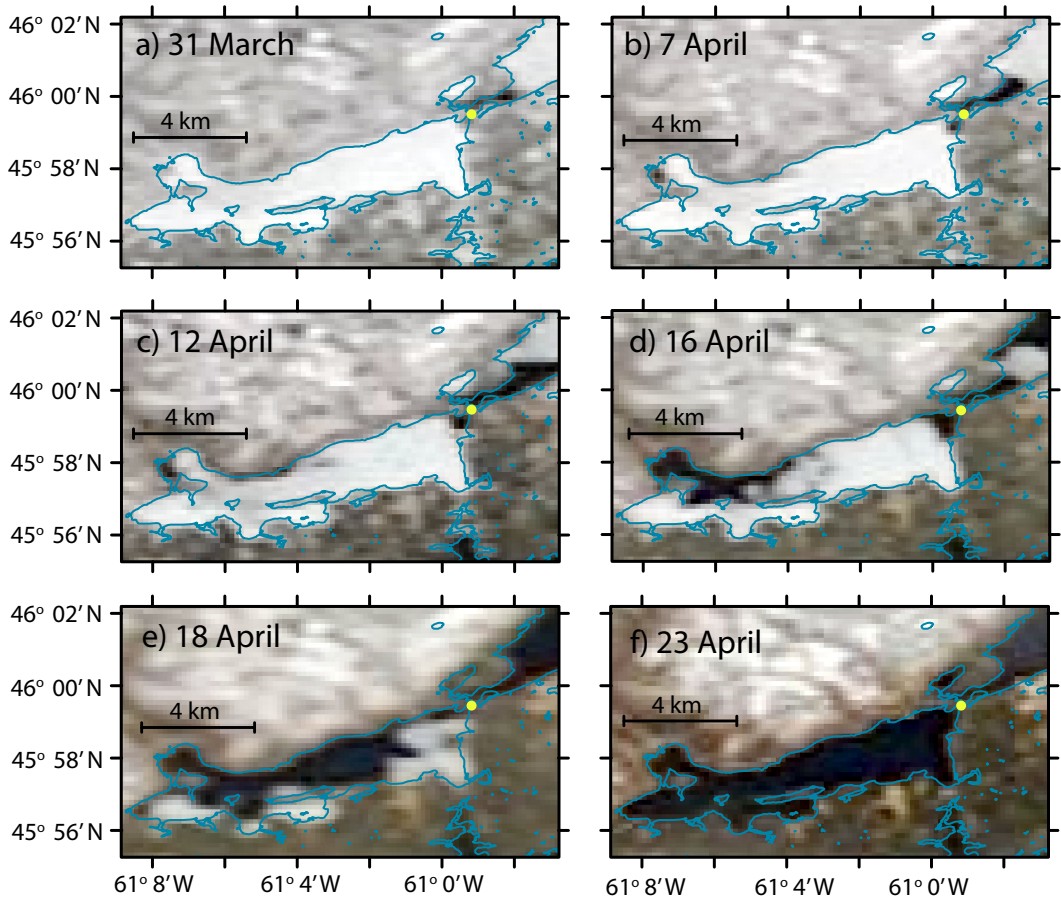

**Figure 2.** MODIS Terra true color images showing changes ice cover during the time-series between 31 March to 23 April. Little Narrows is indicated with a yellow circle on all images. Ice cover retreated most rapidly between 12 April–20 April. Shoreline data (blue lines) is from GeoGratis/Natural Resources Canada (http://geogratis.gc.ca) and satellite data is from NASA Worldview (http://worldview.earthdata.nasa.gov).

## 3.2 Gas transfer velocity

### 3.2.1 Calculation

We calculate the gas transfer velocity using the dual tracer approach (Watson et al., 1991; Wanninkhof et al., 1993). For each experiment, we dissolved a mixture of $^3$He and $SF_6$ in the water, both of which are normally present at very low ambient concentrations, and then measured the change in the ratio of the two gases as a function of time. Measuring two tracers enables correction for dilution and mixing, which reduces the excess concentrations of both gases (relative to air-water equilibrium) but does not change their ratio. Over time, the concentrations of both gases decay toward air-water equilibrium as gas is ventilated

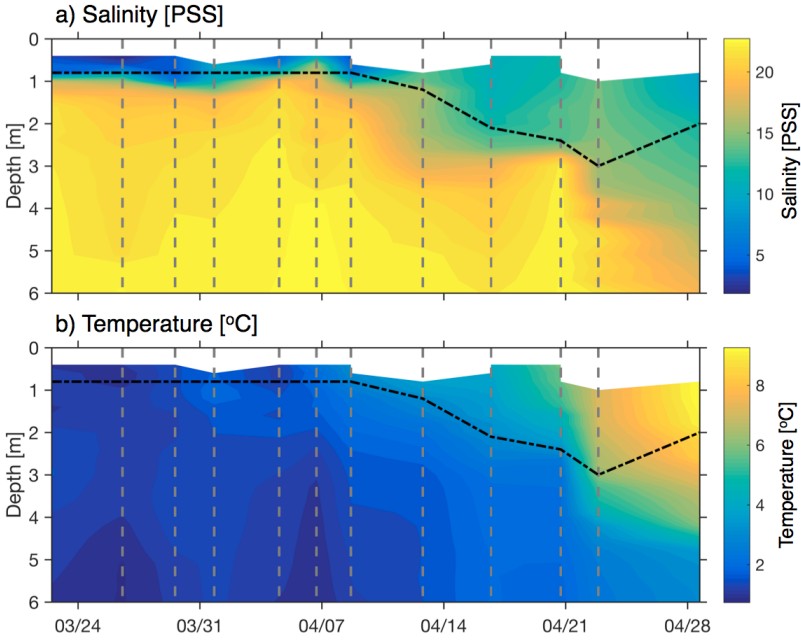

**Figure 3.** (a) Salinity and (b) temperature profiles measured at Little Narrows. The vertical grey dashed lines indicate the timing of CTD casts and the black dot-dashed line shows the mixed layer depth.

to the atmosphere through gas exchange. Because the molecular diffusivity of $^3$He is 8–9 times higher than $SF_6$, $^3$He is lost to the atmosphere more rapidly than $SF_6$ and therefore the $^3$He:$SF_6$ ratio decreases with time. The ratio of the gas transfer velocity for the two gases is expressed as

$$\frac{k_{3He}}{k_{SF_6}} = \left(\frac{Sc_{3He}}{Sc_{SF_6}}\right)^{-n} \tag{1}$$

where $k$ is the gas transfer velocity (m d$^{-1}$) and $Sc$ is the Schmidt number (unitless), defined as the kinematic viscosity of water divided by the molecular diffusivity of the gas in water, and $n$ is an empirical exponent, typically between 0.5–0.67 (Jähne et al., 1984; Liss and Merlivat, 1986). Using a time-series of measurements of the two gases, the gas transfer velocity for $^3$He is calculated as

$$k_{3He} = h\frac{d}{dt}\left(\frac{\ln\left([^3He]_{exc}/[SF_6]_{exc}\right)}{1-(Sc_{SF_6}/Sc_{3He})^{-n}}\right). \tag{2}$$

Here $[^3He]_{exc}$ = $[^3He]_{meas}$ - $[^3He]_{eq}$ where $[^3He]_{exc}$ is the $^3$He excess concentration, $[^3He]_{meas}$ is the measured concentration and $[^3He]_{eq}$ is the equilibrium concentration (calculated as a function of temperature and salinity). $[SF_6]_{exc}$ is defined analogously.

We can write the analytical solution to equation 2 as (Ho et al., 2011b)

$$\left(\frac{[^3He]_{exc}}{[SF_6]_{exc}}\right)_t = \left(\frac{[^3He]_{exc}}{[SF_6]_{exc}}\right)_{t-1}\exp\left(-\frac{k_{3He}\Delta t}{h}\left[1-\left(\frac{Sc_{SF_6}}{Sc_{3He}}\right)^{-n}\right]\right). \tag{3}$$




Using this equation and a cost function, we can find the value of $k_{3_{He}}$ that minimizes the error between the measured and modeled $[^3He]_{exc}/[SF_6]_{exc}$. Once $k_{3_{He}}$ is known, we can calculate $k$ for any other gas using equation 1 by substituting $Sc_{SF_6}$ for the Schmidt number of the gas of interest. For example, for $Sc = 600$ (the Schmidt number for $CO_2$ at 20 °C in freshwater)

$$k_{600} = k_{3_{He}} \left( \frac{600}{Sc_{3_{He}}} \right)^{-n}. \tag{4}$$

5   For this study, we use a Schmidt number dependence of $n = 0.5$ which is appropriate for wavy, unbroken water surfaces (no bubble entrainment) (Jähne et al., 1984; Liss and Merlivat, 1986; Jähne et al., 1987a; Ho et al., 2011b). At Little Narrows, we observed that tidal currents generated surface waves, even at low wind speeds. These waves produce near-surface water turbulence which is the ultimate driver of air-water gas exchange (Jähne et al., 1987b; Wanninkhof et al., 2009).

### 3.2.2   Results

10   The gas transfer velocity was much lower for Injection 1, which was sampled while the basin was essentially full of ice (31 March–10 April), compared to Injection 2, which was sampled when the basin was nearly ice-free (20–23 April). We used equation 3 to model the measurements. We started the model at the time of the first measurement, initialized it with an initial excess concentration ratio and ran it through time for the duration of the injection. We selected the value of $k_{3_{He}}$ yielding the smallest root mean square deviation (RMSD) between the measured ratio and modeled ratio for each injection. The model ran 15   1000 times using a Monte Carlo simulation where the measured excess concentration ratios, including the initial condition, are varied with a Gaussian distribution, with the standard deviation being the estimated measurement error in the ratio (7.3 %).

We assume a constant $Sc_{3_{He}}/Sc_{SF_6}$ and mixed layer depth ($h$) for each injection. In actuality, during each injection $Sc_{3_{He}}$ varies by ∼1 % and the ratio of the Schmidt numbers varies by less than 0.2 %, so this is a small source of error. We assume a mixed layer depth of 0.8(0.3) m for Injection 1. This depth is consistent with the salinity profiles at Little Narrows (Figure 3a) 20   between 31 March and 8 April (between 0.6–1.0 m depth), as well as measurements with a hand-held temperature probe at the site of Injection 1 which indicated that the mixed layer depth was between 0.75–1 m.

For Injection 1, the excess $SF_6$ and $^3He$ concentrations were reduced by two orders of magnitude by the time the tracer reached Little Narrows (7–11 days after injection). The tracer ratio did not display a consistent decrease over the three days we sampled it at Little Narrows, likely in part due to the substantially lower gas transfer velocity. The best fit to all the measure-25   ments was $k_{600} = 0.0457(0.0051)$ m d$^{-1}$, with the uncertainty the standard deviation of the distribution of $k_{600}$ from the Monte Carlo simulation (Figure 4). We avoided collecting discrete tracer samples when the tides were flowing into Whycocomagh Bay, based on visual observation of surface currents, to ensure that the gas transfer velocity reflected the ice coverage within the bay and was not skewed by the open water east of Little Narrows.

The mixed layer appeared to deepen between the CTD profiles on 8 April 16:08 and 12 April 19:13, and it is possible the 30   mixed layer depth on 9–10 April may have been deeper than the estimate of 0.8(0.3) m. However, if this were the case we would expect $k_{600}$ (calculated assuming a constant mixed layer depth) to be lowest when calculated over the longest time period, using the sample collected on 10 April. Instead, the gas transfer velocity was actually the lowest when integrated to 9 April (the excess tracer ratio appears above the best-fit line) and second lowest on 8 April. Since the gas transfer velocities for



**Table 1.** Data for determination of the gas transfer velocity

| | | | Injection 1 | | | |
|---|---|---|---|---|---|---|
| Date | Time | Salinity[a] (PSS) | Temperature[a] (°C) | $\delta^3 He^{[b]}$ (%) | [SF$_6$] (mol L$^{-1}$) | [$^3$He]$_{exc}$/[SF$_6$]$_{exc}$ (mol mol$^{-1}$) |
| 31 March | 15:00 | 0.9 | 2.1 | 17176 | 66.6 | 0.00788 |
| 7 April | 18:00 | 13.2 | 2.6 | 13.8 | 0.152 | 0.00411 |
| 8 April | 17:27 | 8.4 | 2.0 | 56.9 | 0.423 | 0.00480 |
| 9 April | 12:47 | 9.0 | 2.1 | 39.0 | 0.289 | 0.00553 |
| 10 April | 11:02 | 9.3 | 1.8 | 12.6 | 0.115 | 0.00405 |
| | | | Injection 2 | | | |
| Date | Time | Salinity[a] (PSS) | Temperature[a] (°C) | $\delta^3 He^{[b]}$ (%) | [SF$_6$] (mol L$^{-1}$) | [$^3$He]$_{exc}$/[SF$_6$]$_{exc}$ (mol mol$^{-1}$) |
| 20 April | 23:30 | 11.18 | 5.64 | 310.3 | 3.96 | 0.00227 |
| 21 April | 20:00 | 14.91 | 6.49 | 91.8 | 1.60 | 0.00176 |
| 22 April | 12:27 | 14.64 | 7.16 | 43.6 | 0.938 | 0.00150 |
| 23 April | 10:40 | 11.01 | 5.73 | 6.5 | 0.299 | 0.000799 |

[a] During Injection 1, we measured temperature and salinity with a YSI probe to a precision of one decimal place. During Injection 2, we measured temperature with a calibrated thermocouple and salinity with the in situ CTD, to a precision of two decimal places.

[b] $\delta^3$He = (($^3$He/$^4$He)$_{meas}$/($^3$He/$^4$He)$_{eq}$)

Injection 1 integrate over 7–10 days, any change in mixed layer depth during the last 1–2 days will have a small effect on the calculated $k$.

For Injection 2, we use a mixed layer depth of 2.7(0.3) m based on CTD profiles at Little Narrows on 20 April and 23 April, which had mixed layer depths of 2.4 and 3.0 m, respectively (Figure 3a). Calculation of the gas transfer velocity for Injection 2 was relatively straightforward as the ratio of excess $^3$He/SF$_6$ steadily decreased over the five measurements (Table 1). The best fit to all four measurements was $k_{600} = 0.71(0.13)$ m d$^{-1}$ (Figure 4).

Using the published He solubility from Weiss (1971) instead of the unpublished data of Lott and Jenkins (2015, personal communication) results in a gas transfer velocity that is 8 % lower for Injection 1 and 0.4 % lower for Injection 2.

### 3.2.3 Discussion

The gas transfer velocity calculated for Injection 1 is the effective gas transfer velocity ($k_{eff}$); it averages the gas transfer velocity through ice ($k_{ice}$), weighted by the time the tracer spent under ice, and the gas transfer velocity for open water ($k$), weighted by the time the tracer spent in open water (Loose et al., 2014). In partially-ice covered waters, the effective gas



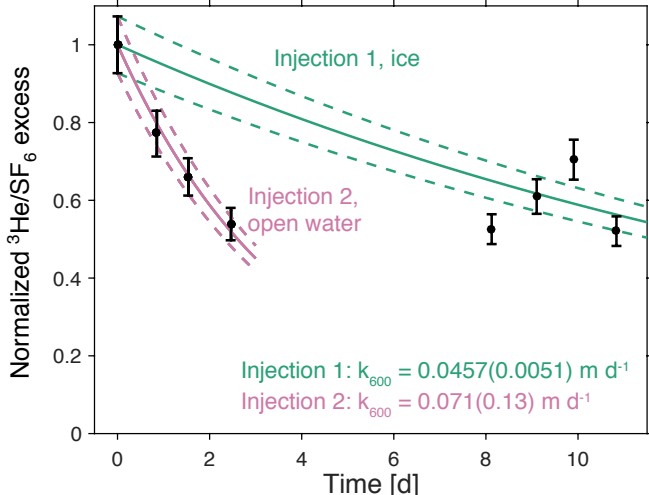

**Figure 4.** Measured and modeled ratio of excess $^3$He/SF$_6$, normalized to the initial measured ratio for each injection. The modeled excess ratio is calculated using the $k_{^3He}$ that minimizes error between the model and measurements. Model results are shown for the model initialized with the initial measured concentration (solid lines), and starting one standard deviation above or below the measured initial concentration (based on an error of 7.3 % in the tracer ratio, dashed lines).

transfer velocity is sometimes calculated as

$$k_{eff} = (f)k + (1-f)k_{ice} \tag{5}$$

where $f$ is the fraction of open water (Loose et al., 2014; Lovely et al., 2015). If $k_{ice}$ is negligible, then $k_{eff} = (f)k$ (Loose and Jenkins, 2014; Crabeck et al., 2014; Butterworth and Miller, 2016). For Injection 2, we determined $k$, the value for open water.

We expect $k_{ice}$ to be lower than $k$ because the ice acts as a physical barrier to gas exchange. The rate of gas molecular diffusion in water (Jähne et al., 1987a; King and Saltzman, 1995) is higher than gas diffusion through ice (Gosink et al., 1976; Ahn et al., 2008; Loose et al., 2011b; Lovely et al., 2015). However, the exact rate of gas diffusion through saltwater ice (and by extension the value of $k_{ice}$) is not well constrained and likely varies based on the physical properties of the ice such as brine volume and temperature (Golden et al., 2007; Loose et al., 2011b; Zhou et al., 2013; Moreau et al., 2014; Lovely et al., 2015). An

additional factor reducing air-water exchange in ice-covered waters is that ice significantly reduces fetch for wave generation and therefore wind-driven near-surface turbulence (Squire et al., 1995; Overeem et al., 2011). However, other processes may enhance near-surface turbulence in the presence of sea ice including convection associated with heat loss and brine rejection (Morison et al., 1992; Smith and Morison, 1993), boundary layer shear between ice and water (McPhee, 1992; Saucier et al., 2004), and wave interactions with drifting ice (Kohout and Meylan, 2008).

To evaluate these results within the framework of equation 5, we must estimate the fractional ice cover during Injection 1. During surveys by car along the shoreline of Whycocomagh Bay and satellite data indicated that the bay was nearly fully covered with a continuous sheet of ice from 31 March–10 April, except for an opening close to Little Narrows (Figure 2a–b).





Beginning between 7 April and 12 April, a small region of water appeared to open up along the shoreline northwest of the site of Injection 1, however, by this time the tracer patch had moved eastward, close to Little Narrows and likely was not significantly affected by this open water (Figure 2c). We mapped out the location of the ice edge closest to Little Narrows by boat on 26 March, 7 April, and 12 April (Figure 1a). Using these surveys and shoreline data, we calculate that for the surface

area of the bay between the injection site and Little Narrows, $f = 0.01$ on 26 March, 0.05 on 7 April, and 0.08 on 12 April. The $f$ experienced by the tracer patch during Injection 1 is likely between 0.05–0.08 because the tracer flowed through the open water near Little Narrows between 6 April - 11 April. If $k$ for open water is the same for both injections, then the results yield $k_{eff} = (f)k$ with $f = 0.064$, which is consistent with the fraction of open water we estimate for Injection 1. Thus we conclude that $k_{ice}$ was negligible, compared to $k$. For example, if $k_{ice}$ was even 10 % of $k$ for open water, then $k_{eff}$ for Injection 1 would

have been $\sim$0.11 m d$^{-1}$, more than double the observed value of 0.0457(0.0051) m d$^{-1}$.

In calculating GOP and NOP by oxygen mass balance, we apply the tracer-based gas transfer velocities estimated by dual tracer release throughout the time-series, since there is no consensus on the best treatment of gas transfer in lakes and estuaries (Clark et al., 1995; Cole and Caraco, 1998; Crusius and Wanninkhof, 2003; Ho et al., 2011a), nor on the parameterization of gas transfer in the presence of ice (Else et al., 2011; Lovely et al., 2015; Butterworth and Miller, 2016). Additionally, if bottom-

derived turbulence (e.g., from tidal flow) is a significant contributor to air-water gas exchange, a parameterization based on wind speed alone may not be appropriate. This method of calculating one average $k_{600}$ for each injection does not enable the development of a wind speed-dependent parameterization for the gas transfer velocity.

Because the $k_{600}$ for Injection 1 and Injection 2 are very different, the treatment of the gas transfer velocity in between the two injections strongly affects the productivity estimates for this period. We use $k_{600} = 0.0457(0.0051)$ m d$^{-1}$ from the

beginning of the time-series until midnight 16 April, the day when Figure 2d was collected. Figure 2d is the first satellite image showing substantial open water within Whycocomagh Bay, but the open water is primarily in the western half of the bay, far from Little Narrows. We use $k_{600} = 0.71(0.13)$ m d$^{-1}$ from midnight 20 April until the end of the time-series on 28 April. Surveys by boat on 19 April and 20 April indicated <10 % ice cover on these days and we collected the first tracer measurements following Injection 2 on 20 April 23:30. Between 16 April and 20 April, we apply a linear interpolation of the

$k_{600}$ for Injection 1 and Injection 2 as a function of time. The gas transfer velocity is most uncertain during the period when the ice cover rapidly decreased because we do not have any measurements of gas transfer at intermediate ice cover. However, because the ice cover retreated rapidly, only four days of the productivity estimates (out of a 33-day time-series) are affected by the uncertainties in gas transfer at intermediate ice cover.

### 3.2.4  Comparison with other estimates

To compare the gas exchange estimates with other published studies, we use wind speed data measured at 10 m height ($u_{10}$) at Eskasoni Reserve, 27 km northeast of Little Narrows (Figure 1) and archived by the Government of Canada (http://climate.weather.gc.ca). The archived data are two-minute averages measured once per hour. For Injection 2, between 20 April 23:00 and 23 April 11:00 the average wind speed was 2.6(1.4) m s$^{-1}$, the median was 2.2 m s$^{-1}$, and the calculated $k_{600}$ over this time period from dual tracer data is 0.71(0.13) m d$^{-1}$. Cole and Caraco (1998) find $k_{600} = 0.636(0.029)$ m d$^{-1}$ (95 % confidence




interval) and their estimate is independent of wind speed in a lake with daily wind speeds of 1.39(0.06) m s$^{-1}$ (95 % confidence interval); this $k_{600}$ is consistent within uncertainty with the Injection 2 result. Standard open ocean parameterizations that use a quadratic dependence on wind speed predict $k_{600} \simeq$ 0.5–0.6 m d$^{-1}$ with uncertainties of ∼20 % or ∼0.10 m d$^{-1}$ (Ho et al., 2006; Sweeney et al., 2007; Wanninkhof, 2014).

Crusius and Wanninkhof (2003) found that in a lake gas exchange can be estimated nearly equally well using three different parameterizations. Using their parameterizations with the wind speed record during our time-series, we calculate transfer velocities of 0.32–0.66 m d$^{-1}$ and the velocity is most similar to our result when we use a constant gas transfer velocity $k_{600}$ = 0.24 m d$^{-1}$ for $u_{10} <$ 3.7 m s$^{-1}$ and $k_{600} = 1.23 u_{10} - 4.30$ for $u_{10} \geq$ 3.7 m s$^{-1}$. However, Crusius and Wanninkhof (2003) parameterized the gas transfer using instantaneous (e.g. one-minute averaged) winds measured throughout the time-series, not

once per hour, and emphasize the importance of including the variability in short-term winds when quantifying gas exchange at low wind speeds. If gas transfer velocity has a nonlinear dependence on wind speed, then short-term measurements will more accurately represent the gas transfer than wind speeds averaged over longer periods (Livingstone and Imboden, 1993; Crusius and Wanninkhof, 2003). Since we only have two-minute averages measured once per hour (for a total of 60 measurements during Injection 2) the wind record we use may not fully represent the variability in winds during the Injection 2 measurement

period.

    A source of error in comparisons with published results is that the wind speed data come from a different location than the study area. Although Eskasoni Reserve is adjacent to the Bras d'Or Lake, the local topography and bathymetry is different near the reserve and in Whycocomagh Bay. Thus, it is likely that the wind speed and momentum stress at the air-water interface differs at Whycocomagh Bay compared to Eskasoni Reserve (Ortiz-Suslow et al., 2015).

The measurements are in agreement with other studies showing gas transfer velocity is significantly reduced under near-complete ice cover (Lovely et al., 2015; Butterworth and Miller, 2016) and contrast with studies showing enhanced gas transfer under >85 % ice cover (Fanning and Torres, 1991; Else et al., 2011). We find that $k_{eff} = (f)k$ for >90 % ice cover but we cannot evaluate whether the same equation holds at intermediate ice cover because there was no injection at a lower fractional ice cover. In this study, the ice cover was near-continuous across the entire bay during Injection 1 and likely did not contain the polynyas

and leads that are prevalent in the Arctic and Antarctic; differences in gas transfer behaviour are expected based on the nature of the ice pack.

### 3.3   Gross oxygen production

#### 3.3.1   Calculation

The triple oxygen isotopic composition of $O_2$ is an effective tracer of gross photosynthetic $O_2$ production (Juranek and Quay,

2013). Due to reactions in the upper atmosphere that impart a small mass-independent isotopic signature on atmospheric oxygen, $O_2$ derived from air-water exchange (from the atmosphere) has a unique triple isotopic signature compared to $O_2$ generated by photosynthesis (from $H_2O$) and $O_2$ consumed by respiration. We characterize the oxygen isotopic composition





using

$$\delta^{18}O = X^{18}/X^{18}_{std} - 1, \tag{6}$$

and express the $\delta^{18}O$ in ‰ by multiplying by 1000. Here $X^{18} = r(^{18}O/^{16}O)$ is the measured ratio and $X^{18}_{std} = r(^{18}O/^{16}O)_{std}$ is the ratio of the isotopes in the standard. We calculate $\delta^{17}O$ analogously. For GOP studies, $O_2$ in air is the standard for isotopic

measurements of $O_2$ and $H_2O$ is referenced to the VSMOW-SLAP scale. For clarity, we distinguish between the isotopic composition of the two substrates ($O_2$ and $H_2O$) as $\delta^{18}O$-$O_2$ and $\delta^{18}O$-$H_2O$.

The term $^{17}\Delta$ quantifies the triple isotopic composition of dissolved $O_2$

$$^{17}\Delta = \ln(\delta^{17}O\text{-}O_2 + 1) - \lambda \ln(\delta^{18}O\text{-}O_2 + 1). \tag{7}$$

We report $^{17}\Delta$ with $\lambda = 0.5179$, the ratio of the fractionation factors for respiratory $O_2$ consumption in $^{17}O$ relative to $^{18}O$

(i.e. $\lambda = {}^{17}\varepsilon/{}^{18}\varepsilon$, where $\varepsilon$ is the isotopic fractionation of $O_2$ due to respiratory consumption) (Luz and Barkan, 2005). This value for $\lambda$ is selected so that $^{17}\Delta$ is nearly unaffected by respiratory $O_2$ consumption and reflects the proportion of $O_2$ that is derived from photosynthesis relative to air-water gas exchange (Hendricks et al., 2005; Juranek and Quay, 2013; Nicholson et al., 2014). We report $^{17}\Delta$ in per meg (1 per meg = 0.001 ‰) due to the small range of values (typically 8 – 294 per meg) in natural waters.

Two key constraints in the calculation of GOP from measurements of the triple isotopic composition of $O_2$ are the isotopic composition of $O_2$ derived from air-water exchange, and the isotopic composition of photosynthetic $O_2$. The composition of photosynthetic $O_2$ is dependent on the triple oxygen isotopic composition of $H_2O$, the substrate for photosynthetic $O_2$, and the isotopic fractionation associated with photosynthetic $O_2$ production. In oceanic studies, a common assumption is that the $H_2O$ isotopic composition is equivalent to VSMOW (standard mean ocean water), but in brackish systems it is necessary to estimate

the isotopic composition of water and incorporate this into the GOP calculation Manning et al. (2017a).

Because we did not measure the triple oxygen isotopic composition of $H_2O$, we use previously published measurements of $\delta^{18}O$-$H_2O$ and published relationships between $\delta^{17}O$-$H_2O$ and $\delta^{18}O$-$H_2O$ to estimate the values of $\delta^{18}O$-$H_2O$ and $\delta^{17}O$-$H_2O$ during the time-series, as described in Manning et al. (2017a). We assume that the waters in the estuary represent a mixture of two endmembers: seawater and meteoric (precipitation-derived). We define the salinity and $\delta^{18}O$-$H_2O$ for the two endmembers

and then calculate $\delta^{18}O$-$H_2O$ for each water sample collected during the time-series as a linear function of salinity A similar approach is applied for $\delta^{17}O$-$H_2O$.

For the seawater endmember, we use compilations of $\delta^{18}O$-$H_2O$ and salinity (Schmidt, 1999; Bigg and Rohling, 2000) available from an online database (Schmidt et al., 1999). We included all 19 near-surface samples ($<5$ m depth) between 44–48 °N and 58–64 °W in the database. For these samples, the average $\delta^{18}O$-$H_2O$ = -1.68(0.26) ‰ and salinity = 31.25(0.30)

PSS.

For the meteoric endmember, we use an 8-year time-series of $\delta^{18}O$-$H_2O$ measured in Truro, Nova Scotia (200 km southwest of our study area, 40 m elevation) and archived in the Global Network of Isotopes in Precipitation (GNIP) database (IAEA/WMO, 2016). The amount-weighted value of $\delta^{18}O$-$H_2O$ over the time-series was -9.3(3.1) ‰ versus VSMOW, using precipitation measurements from Truro NS over the same time period from the Government of Canada historical weather





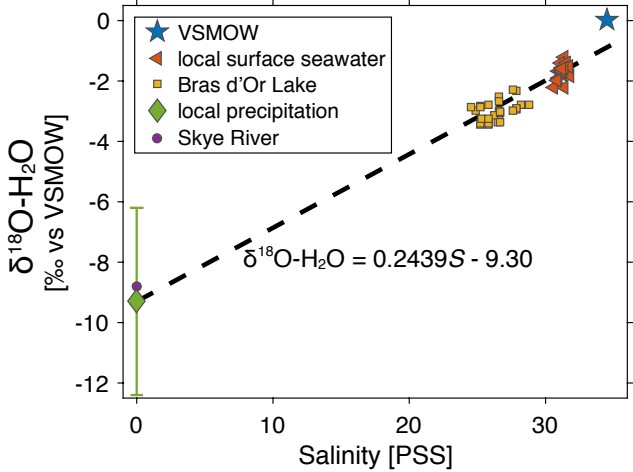

**Figure 5.** Measurements of $\delta^{18}$O-H$_2$O in local surface seawater, the Bras d'Or Lake, local precipitation, and Skye River (within the Why-cocomagh Bay watershed). The regression is calculated using the local precipitation and seawater values. The local precipitation value is plotted as 9.3(3.1) ‰, with the error bar the standard deviation of the amount-weighted annual average.

database (http://climate.weather.gc.ca). Also, Timsic and Patterson (2014) measured $\delta^{18}$O-H$_2$O = -8.8(0.1) ‰ on a water sam-
ple collected in July 2009 from the Skye River, within the Whycocomagh Bay watershed (Parker et al., 2007), consistent with
the average Truro, NS value.

Using the two endmembers, S = 0 PSS, $\delta^{18}$O-H$_2$O = -9.3 ‰ (local meteoric water) and S = 31.25 PSS, $\delta^{18}$O-H$_2$O $= -1.68‰$

(local seawater), we derive the equation $\delta^{18}$O-H$_2$O = 0.2439$S$ - 9.30 (Figure 5). This equation is consistent with published
$\delta^{18}$O-H$_2$O measurements from within the estuary (Figure 5). Mucci and Page (1987) collected water samples from the Bras
d'Or Lake in November 1985 and found a salinity of S = 26.42(1.12) PSS and $\delta^{18}$O = -2.99(0.32) ‰ for samples at 17 different
stations (albeit none within Whycocomagh Bay). Notably, VSMOW (S = 34.5 PSS) plots 0.9 ‰ above the local mixing line,
which demonstrates the importance of accurately defining both the freshwater and seawater endmembers.

Then, for the two endmember values of $\delta^{18}$O-H$_2$O, we estimate $\delta^{17}$O-H$_2$O using the following equation:

$$^{17}\text{O-excess} = \ln(\delta^{17}\text{O-H}_2\text{O} + 1) - \lambda_w \ln(\delta^{18}\text{O-H}_2\text{O} + 1) \tag{8}$$

with $\lambda_w$ = 0.528 and all isotopic compositions referenced to VSMOW-SLAP. The value of $\lambda_w$ = 0.528 is well established for
meteoric waters and seawater (Meijer and Li, 1998; Landais et al., 2008; Luz and Barkan, 2010). Spatial variability in the
$^{17}$O-excess of natural waters is less well understood due to the currently limited observations at sufficient accuracy to resolve

the small excess (Luz and Barkan, 2010; Li et al., 2015). To calculate the freshwater and seawater endmembers for $\delta^{17}$O-H$_2$O
we use the average values of $^{17}$O-excess of 33 per meg for meteoric water and -5 per meg for seawater (Luz and Barkan,
2010). The endmembers are $\delta^{17}$O-H$_2$O = -4.888 ‰ and -0.908 ‰ for meteoric water and seawater respectively, and the linear
regression is $\delta^{17}$O-H$_2$O = 0.1274$S$ - 4.89. These $\delta$ values for H$_2$O, referenced to VSMOW, are subsequently referenced to
atmospheric O$_2$ using results from Barkan and Luz (2011). In this study, the choice of $\delta^{18}$O-H$_2$O is more important than the





$^{17}$O-excess because $^{17}$O-excess varies by less than 0.1 ‰ between samples whereas the freshwater $\delta^{18}$O-H$_2$O differs from VSMOW by 9.1 ‰. We discuss the sensitivity of the GOP calculations to the assumed $^{17}$O-excess and $\delta^{18}$O-H$_2$O below, and the effect of other processes on the isotopic composition of H$_2$O in section 3.6.

We calculate GOP using equation 7 from Manning et al. (2017a)

$$
\quad \mathrm{GOP} = k_{O_2}[\mathrm{O_2}]_{eq} \frac{\left[\frac{X^{17}-X_{eq}^{17}}{X^{17}} - \lambda \frac{X^{18}-X_{eq}^{18}}{X^{18}}\right]}{\left[\frac{X_p^{17}-X^{17}}{X^{17}} - \lambda \frac{X_p^{18}-X^{18}}{X^{18}}\right]} + \frac{h[\mathrm{O_2}]\frac{\partial^{17}\Delta}{\partial t}}{\left[\frac{X_p^{17}-X^{17}}{X^{17}} - \lambda \frac{X_p^{18}-X^{18}}{X^{18}}\right]}. \tag{9}
$$

In this equation, $k_{O_2}$ is the gas transfer velocity for O$_2$ (m d$^{-1}$), [O$_2$] is the O$_2$ concentration (mol m$^{-3}$), $h$ is the mixed layer depth (m), $X^{17} = r(^{17}\mathrm{O}/^{16}\mathrm{O})$, and $\lambda = 0.5179$ (equation 7). The subscripts $eq$ and $p$ refer to O$_2$ at air-water equilibrium and produced by photosynthesis and terms without a subscript ([O$_2$], $X^{17}$, and $^{17}\Delta$) are the measured mixed layer values. The first term on the right side of equation 9 is the steady state GOP term, and the second term is the non-steady state GOP term. If the

system is at steady state with respect to $^{17}\Delta$ (i.e., there is no change in $^{17}\Delta$ with time) then the second term on the right side of equation 9 equals zero and can be eliminated.

We calculate $X_{eq}^{18}$ based on Benson and Krause (1980a, 1984), and $X_{eq}^{17}$ using $^{17}\Delta_{eq} = 8$ per meg (Reuer et al., 2007; Stanley et al., 2010), which is consistent with the daily measurements of distilled water equilibrated at room temperature that were analyzed along with the environmental samples (8.1 per meg with standard error of 1.6 per meg, $n = 12$), as well as prior

measurements of distilled water equilibrated at <5 °C (R.H.R. Stanley, unpublished data). We calculate $X_p^{18}$ and $X_p^{17}$ using the salinity-dependent isotopic composition of H$_2$O defined above, and isotopic fractionation factors for photosynthetic O$_2$ with respect to H$_2$O based on data in Luz and Barkan (2011) for average phytoplankton. The Matlab code used to calculate GOP and the triple oxygen isotopic composition of water (from two-endmember mixing of $\delta^{18}$O-H$_2$O and salinity) is available online (Manning and Howard, 2017).

We calculate gross oxygen production using samples collected at Little Narrows from 25 March–27 April (Figure 6). Visual inspection of the $^{17}\Delta$ data indicated that $^{17}\Delta$ changed during the time-series and therefore the calculation includes a non-steady state GOP term. The non-steady state term in equation 9 is $h[\mathrm{O_2}]\partial^{17}\Delta/\partial t$. To calculate the rate of change in $^{17}\Delta$ with time, we first averaged the data into 24-hour bins (beginning and ending at 19:30, local sunset) to avoid over-weighting times when samples were collected at higher frequency. We calculated the average $^{17}\Delta$ and sampling time for all samples collected each

day. Next, we separated the data into two periods: one period began on 25 March and ended 19 April 07:30, and the second period covered the remainder of the time-series (ending 27 April). A linear regression of $^{17}\Delta$ versus time for the two time periods yielded a slope of 0.67 per meg d$^{-1}$ ($r^2 = 0.47$) for the first period and -2.99 per meg d$^{-1}$ ($r^2 = 0.94$) for the second period. The approximate timing for the change between periods was determined by visual inspection and then adjusted to maximize the $r^2$ and so that the equations of the two lines gave very similar $^{17}\Delta$ values at 19 April 00:00 (within 1 per meg).

Splitting the period from 25 March–19 April into two separate regressions (or one period where $^{17}\Delta$ increased and one period where it was constant) yielded much lower $r^2$ values and a discontinuous $^{17}\Delta$ record (different $^{17}\Delta$ values at the end of one period and the start of another), so a single regression was used for this period.





**Figure 6.** Gross oxygen production at Little Narrows and data used in the calculation. $^{17}\Delta$ measurements from (a) all samples and (b) daily averaged values with the two linear regressions shown. Gross oxygen production in (c) mixed layer-integrated and (d) volumetric units. The green diamonds and blue squares show the values of the two terms in the GOP calculation (steady state, SS and non-steady state, NSS), and the yellow circles show the total GOP (the sum of the two terms). Error bars are only shown on the total GOP for clarity. The SS and NSS terms are slightly offset in time to make it easier to see both terms at the start of the time-series. (e) cumulative GOP in mixed layer-integrated and volumetric units; error (pink shaded range of values) is only shown for the volumetric GOP for clarity. (f) gas transfer velocity for a gas with a Schmidt number of 600 (blue line), and for $O_2$ (yellow circles), which has a variable Schmidt number based on temperature and salinity). The grey shaded area in all plots is the period where gas exchange and the non-steady state GOP term are most uncertain.





The other two variables in the non-steady state GOP term are the mixed layer depth ($h$) and $[O_2]$. We estimate $[O_2]$ as

$$[O_2] \simeq [O_2]_{eq} \frac{O_2/Ar}{O_2/Ar_{eq}}. \qquad (10)$$

This estimate assumes that $[Ar] = [Ar]_{eq}$. If $[Ar]$ is, for example, 2 % supersaturated then the estimated $[O_2]$ and non-steady state GOP term will be 2 % too high (Cassar et al., 2011).

Using equation 9, GOP is estimated for each sample, using an isotopic composition for $H_2O$ based on the salinity of the sample, and a mixed layer depth, rate of change in $^{17}\Delta$ with time, and gas transfer velocity based on the sampling time, and then calculate the daily average GOP from all samples on a given day (beginning and ending at 19:30, local sunset). A few days had no measurements and the remainder of days had between 1–4 samples. The uncertainty in GOP on each day is calculated by propagating uncertainty in $k_{O_2}$ (11 % for Injection 1, 18 % for Injection 2), uncertainty in the mixed layer depth (from

10–38 %, 0.3 m), uncertainty in the rate of change in $^{17}\Delta$ with time (22 % and 9 % where $^{17}\Delta$ is increasing and decreasing, respectively), and uncertainty in the photosynthetic endmember (discussed below). Measurement uncertainty in the isotopic composition of $O_2$ is excluded from the error calculation because it is a random error, rather than a systematic error (meaning that by taking many measurements of $^{17}\Delta$ over several days, the measurements with high and low $^{17}\Delta$ will average out) and because the measurement error is smaller than most other sources of error. All uncertainties are expressed as the standard

deviation.

The isotopic composition of $H_2O$ one of the largest sources of error: if the $^{18}O$-$H_2O$ endmembers for meteoric water and local seawater are changed to the minimum values of -12.4 ‰ and -1.94 ‰ (one standard deviation below the mean value) and then $\delta^{18}O$-$H_2O$ and $\delta^{17}O$-$H_2O$ are re-calculated for each sample, GOP is on average 48 % higher. If we shift the $\delta^{18}O$-$H_2O$ endmembers for meteoric water and local seawater to the maximum values of -6.2 ‰ and 1.42 ‰, respectively, GOP is on

average 23 % lower. The calculated GOP increases nonlinearly as the isotopic composition of photosynthetic $O_2$ becomes more different from the isotopic composition of equilibrated $O_2$ (Manning et al., 2017a). If the $^{17}O$-excess of $H_2O$ is increased or decreased by 20 per meg, GOP changes by an average of 10 %.

GOP calculated with the local isotopic composition of $H_2O$ is 46–97 % higher (mean 74 %) than GOP calculated assuming the water's isotopic composition is equivalent to VSMOW. Using the local isotopic composition of water instead of VSMOW

is particularly important in this study because the system is not pure seawater. However, even in some oceanic regions such as the Arctic, the isotopic composition of $H_2O$ can be substantially different from VSMOW (LeGrande and Schmidt, 2006). The definition and importance of the photosynthetic endmember for GOP calculations in different environments warrants further review (Manning et al., 2017a).

We calculate the mixed layer-integrated GOP (mmol $O_2$ m$^{-2}$ d$^{-1}$) and the volumetric GOP (mmol $O_2$ m$^{-3}$ d$^{-1}$), which is

the mixed layer-integrated GOP divided by the mixed layer depth. For this time-series GOP is only calculated for the mixed layer because there are no $O_2$ measurements below the mixed layer. The average errors in the daily GOP are $^{+77}_{-49}$ % and $^{+52}_{-31}$ % for the volumetric and mixed layer-integrated GOP, respectively. We also calculate the cumulative GOP (the sum of the daily GOP estimates, from the start of the time-series until a given day) which provides an estimate of the total amount of photosynthetic $O_2$ produced by autotrophs during our time-series (Figure 6e) (Fassbender et al., 2016).



### 3.3.2 Results and discussion

GOP is most uncertain from 16 April through 19 April (grey shaded area in Figure 6). During this period, the ice cover was retreating and we do not have an estimate of $k_{O_2}$ at intermediate ice cover, nor do we have estimates of ice cover on 17 April or 19 April, because clouds obscured the satellite images on those days. Additionally, the exact timing of the change in the sign

of the non-steady state term is unclear. For example, in Figure 6b, 18 April is the final day where $\partial^{17}\Delta/\partial t$ is positive and the calculated GOP is 58 mmol $O_2$ $m^{-2}$ $d^{-1}$. If 18 April is included in the period where $^{17}\Delta$ is decreasing, the calculated GOP is 27 mmol $O_2$ $m^{-2}$ $d^{-1}$ on 18 April. The concentration of photosynthetic $O_2$ is highest during this time ($^{17}\Delta$ = 48–54 per meg on 16 April, 18 April, and 19 April), despite the increasing gas transfer velocity. Therefore it is likely that GOP peaked at the end of the ice melt period and then declined to values similar to the beginning of the time-series, but the uncertainties in GOP

from 16–19 April are large.

Overall, the rate of volumetric mixed layer GOP was relatively constant throughout the time-series (excluding the period from 16 April–20 April, when values are most uncertain) at $5.5(^{+2.9}_{-1.7})$ mmol $O_2$ $m^{-3}$ $d^{-1}$. The mixed layer-integrated GOP showed larger changes with time that are related to the influence of changes in the mixed layer depth on the non-steady state term. From 25 March through 8 April, mixed layer-integrated GOP was $4.6(^{+3.7}_{-2.6})$ mmol $O_2$ $m^{-2}$ $d^{-1}$. Beginning after 8 April,

the mixed layer depth began to increase and the non-steady state calculation showed a substantial increase in mixed layer-integrated GOP, to $7.2(^{+5.4}_{-3.3})$ mmol $O_2$ $m^{-3}$ $d^{-1}$ on 14 April, a 56 % increase. The non-steady state GOP term is multiplied by the mixed layer depth, and therefore it increases linearly as the mixed layer deepens, causing the total mixed layer-integrated GOP to decrease. However, the non-steady state GOP term is constant in volumetric units because the mixed layer depth cancels out of the equation. After the ice is gone, $^{17}\Delta$ begins to decrease and so does the mixed layer-integrated GOP. On the

last four days of the time-series, the mixed layer-integrated mixed layer GOP is $8.7(^{+3.6}_{-6.3})$ mmol $O_2$ $m^{-2}$ $d^{-1}$, 89 % higher than the average value prior to 9 April, but the mixed layer is also more than twice as deep at the end of the time-series compared to the beginning.

The influence of mixed layer depth on the GOP calculations is demonstrated with the cumulative GOP (Figure 6e). The volumetric GOP is roughly constant with time and therefore the cumulative volumetric GOP increases in an approximately

linear fashion, except during the period from 16–19 April, where GOP is most uncertain. The mixed layer-integrated GOP is slightly lower than the volumetric GOP at the start of the time-series (because the mixed layer is slightly less than 1 m deep) and then as the mixed layer deepens, the cumulative mixed layer-integrated GOP increases rapidly. In the last few days of the time-series the mixed layer shoals and the rate of increase in cumulative mixed layer-integrated GOP is lower.

Overall, the result that volumetric GOP was similar at the beginning and end of the time-series indicates that ice-free

conditions are not a pre-requisite for phytoplankton growth in this system. Currently, ecosystem dynamics within and below ice formed from fresh and brackish waters are not well understood (Salonen et al., 2009; Bertilsson et al., 2013; Hampton et al., 2015). Other investigators have shown that photosynthetic microbes can inhabit the interior, upper surface, and lower surface of ice, and tend to be most concentrated on the bottom surface (Welch et al., 1988; Cota et al., 1991; Frenette et al., 2008; Boetius et al., 2013). Traditionally, investigators have argued that ice-associated communities are most prevalent in ice formed from



seawater; as salinity increases, the volume of unfrozen brines within the ice that the microbes can inhabit increases, and the bottom surface of the ice becomes more uneven, increasing bottom algal settlement efficiency (Legendre et al., 1981; Gosselin et al., 1986). However, more recently, investigators have also found algae growing within and on the bottom of freshwater ice in lakes and rivers, including locations in Canada such as the Great Lakes and the St. Lawrence River (Bondarenko et al., 2006;

Frenette et al., 2008; Twiss et al., 2012; D'souza et al., 2013).

Phytoplankton can also grow in the water column beneath ice, especially thinner first-year ice (Legendre et al., 1981; Mundy et al., 2009; Arrigo et al., 2012). Bare ice transmits more light to surface waters than snow-covered ice, and melt pond-covered ice transmits four times as much light as bare ice (Light et al., 2008; Arrigo et al., 2012; Light et al., 2015). First-year ice in the Arctic (0.5–1.5 m thick) transmits ∼47–75 % of incident light through melt pond-covered ice and ∼13–25 % of incident

light through snow-free ice (Arrigo et al., 2012; Light et al., 2015). The ice in the Bras d'Or Lake near the site of Injection 1 was ∼0.3 m thick on 29 March and therefore likely similar or greater fractions of light were transmitted through the ice. Ice transmitting just 2 % of surface irradiance may support high rates of photosynthetic activity, if the microbes are acclimated to lower light levels (Cota, 1985). We observed melt ponds on the Bras d'Or Lake during tracer injections on 31 March and frequently during visual surveys in April. The shallow mixed layer prior to ice melt (∼0.8 m from the beginning of the time-

series until 8 April) would have kept phytoplankton in the mixed layer close to the surface and therefore receiving light that penetrated through the ice.

Our $O_2$ mass balance techniques will record GOP by free-floating phytoplankton in the water column below the ice, as well as GOP by ice-associated phytoplankton if the $O_2$ they produce diffuses into the water rather than into the ice surface or atmosphere. Bottom-associated algae likely release much of their $O_2$ into the water column, especially for filamentous forms

such as the diatoms frequently observed in Lake Erie and the Arctic (D'souza et al., 2013; Boetius et al., 2013).

### 3.4 Net oxygen production and export efficiency

#### 3.4.1 Calculation

We quantify non-steady state NOP, incorporating the observed changes in $O_2/Ar$ during the time-series. We estimate NOP as

$$\text{NOP} = k_{O_2}\Delta(O_2/Ar)[O_2]_{eq} + h\frac{\partial\Delta(O_2/Ar)}{\partial t}[O_2]_{eq} \tag{11}$$

where $k_{O_2}$ is the real-time gas transfer velocity (m d$^{-1}$), $[O_2]_{eq}$ is the equilibrium $O_2$ concentration (mol m$^{-3}$) and $h$ is the mixed layer depth (m) (Hamme et al., 2012). The first term on the right side of equation 11 is the steady state NOP term, and the second term is the non-steady state NOP term, which is dependent on the rate of change in $\Delta(O_2/Ar)$ with time. To calculate the rate of change in $\Delta(O_2/Ar)$ with time, we resampled the data to a fixed 5 s interval (each scan of all masses took 5–6 s) and filled in gaps with a linear interpolation. Then we applied a third order lowpass Butterworth filter with a cutoff frequency of 0.3

d$^{-1}$ to generate a smooth $O_2/Ar$ record (Roberts and Roberts, 1978) (Figure 7). We selected the cutoff frequency to remove the short-term variability from tides and diel changes in photosynthesis and respiration, and to minimize the number of times the inflection of the curve changed while capturing the overall trends in $O_2/Ar$. Below we discuss the sensitivity of the calculated



NOP to the choice of cutoff frequency. Finally, we calculated the derivative of $\Delta(O_2/Ar)$ with respect to time using the filtered record. We applied the same filtering method to the in situ salinity and thermocouple temperature data and used the filtered data to calculate the $[O_2]_{eq}$ and $k_{O_2}$, to prevent short-term fluctuations in salinity and temperature from producing apparent changes in NOP. We calculate the daily NOP (from 19:30 to 19:29 local time) using the average $\partial\Delta(O_2/Ar)/\partial t$ based on the

filtered record and the average $\Delta(O_2/Ar)$ (using the raw, unflitered data).

To calculate the uncertainty in NOP, uncertainty in $k_{O_2}$ (11 % for Injection 1, 18 % for Injection 2), mixed layer depth (0.3 m, 10–38 %), and the non-steady state term are propagated. Uncertanty in the non-steady state term is based on the results using different filtering methods. For the cumulative NOP over the whole time-series, the non-steady state NOP changes by 3–13 % if we double or halve the cutoff frequency. A conservative 13 % error associated with the cutoff frequency choice is

included in the estimates of daily and cumulative NOP. Uncertainty in $\Delta(O_2/Ar)$ (<0.1 %, based on the mean offset between the EIMS and the discrete samples) has a negligible impact on NOP, relative to the other sources of error. The average errors in the daily NOP are $\pm$ 0.8 mmol $O_2$ m$^{-2}$ d$^{-1}$ (34 %) and $\pm$0.3 mmol $O_2$ m$^{-3}$ d$^{-1}$ (23 %) for the mixed layer-integrated and volumetric NOP, respectively. All uncertainties are the standard deviation.

Finally, we calculate the ratio of NOP to GOP for each daily estimate (the export efficiency). This ratio is similar to an f-ratio

or an e-ratio (Dugdale and Goering, 1967; Laws et al., 2000) and provides information on the fraction of GOP that is available for export out of the mixed layer (Figure 7f). The uncertainties in the NOP/GOP ratio on each day are quite large. In a steady state NOP and GOP calculation, the gas transfer velocity $k_{O_2}$ cancels out of the equation for the NOP/GOP ratio and therefore it is not a source of uncertainty; however, in the non-steady state term the $k_{O_2}$ does not cancel out.

### 3.4.2   NOP results and comparison of of NOP and GOP

Based on the non-steady state NOP estimates, the ecosystem was on average net autotrophic as the ice was melting, from the beginning of the time-series through 15 April (mean volumetric NOP of 1.9(2.1) mmol $O_2$ m$^{-3}$ d$^{-1}$, median 2.5 mmol $O_2$ m$^{-3}$ d$^{-1}$). During the (nearly) ice-free period from 20 April through the end of the time-series, the community was on average net heterotrophic but with a smaller magnitude than during the start of the time-series (mean volumetric NOP of -0.7(0.9) mmol $O_2$ m$^{-3}$ d$^{-1}$, median -0.7 mmol $O_2$ m$^{-3}$ d$^{-1}$). When the bay was nearly full of ice cover (from the beginning of the

time-series until $\sim$16 April), NOP was dominated by the non-steady state term and this term was positive except for between 31 March–3 April when it was negative but small in magnitude. As the ice cover decreased, the non-steady state term decreased and became negative on 18 April. From 18–22 April the steady state term is roughly equal in magnitude but opposite in sign to the non-steady state NOP term. The NOP is more strongly negative from 23–25 April (volumetric NOP of -1.6(0.5) mmol $O_2$ m$^{-3}$ d$^{-1}$) and then on the last two days of the time-series $\Delta(O_2/Ar)$ was close to 0 and so was the rate of change in $\Delta(O_2/Ar)$

with time (volumetric NOP of -0.7 mmol $O_2$ m$^{-3}$ d$^{-1}$). If the time-series had continued for longer, it would have been possible to observe whether the NOP value eventually stabilized near 0 following the dynamic ice melt period, or if it continued to oscillate between periods of net autotrophy and net heterotrophy.

Similarly to GOP, NOP is most uncertain between 16–19 April due to uncertainties in the fractional ice cover and the parameterization of gas exchange. However, the rate of change with time term is more certain for NOP during this period,



**Figure 7.** Net oxygen production at Little Narrows and data used in the calculation. (a) Δ(O$_2$/Ar) and (b) temperature measurements. The blue lines are the raw data and the black line is the filtered data. NOP in (c) mixed layer-integrated and (d) volumetric units. The yellow diamonds and pink triangles show the values of the two terms in the GOP calculation (steady state, SS and non-steady state, NSS). (e) cumulative NOP in mixed layer-integrated and volumetric units. (f) export efficiency ratio (NOP/GOP). The shaded grey area is the period where ice cover was decreasing rapidly and NOP is most uncertain.




compared to the same period in the GOP calculation, because the high-frequency measurements allowed us to calculate a smoothly varying non-steady state term, rather than an abrupt change in the non-steady state term based on a linear fit. NOP appears to be decrease during the period where the ice cover changes most rapidly.

When summed over the entire time-series, the magnitude of the cumulative volumetric NOP (37.6(6.3) mmol $O_2$ m$^{-3}$) is greater than the cumulative mixed layer-integrated NOP (28.8(4.4) mmol $O_2$ m$^{-2}$). Volumetric NOP is negative in the second half of the time-series and the mixed layer is deepest during this period, causing the cumulative mixed layer-integrated NOP to decrease substantially. The cumulative NOP contrasts with cumulative GOP; the mixed layer-integrated GOP exceeds the volumetric GOP because GOP is always positive and the mixed layer was on average > 1 m deep.

The ratio of NOP/GOP has large uncertainties but qualitatively follows the trends of NOP, since GOP was relatively constant throughout the time-series (except during and immediately following the loss of ice cover). At the start of the time-series, there are two dates where NOP/GOP > 1 which by definition is not possible. The high NOP/GOP values could be due to uncertainty in the isotopic composition of water, which enters into the GOP calculation but not the NOP calculation, and/or the non-steady state terms for GOP and NOP. Vertical mixing is another possible cause of the estimated NOP/GOP ratios exceeding 1; because we do not have measurements below the mixed layer, we cannot compute a correction for vertical mixing. The gradients in $^{17}\Delta$ and $\Delta(O_2/Ar)$ are likely different with depth, leading to different magnitudes of the vertical mixing correction for NOP and GOP, which could potentially lead to errors in the estimated NOP/GOP ratio (section 3.6).

The different trends in GOP and NOP warrant some discussion. Volumetric GOP was relatively constant throughout the time-series except briefly at the end of the ice melt period where it was likely somewhat higher, whereas NOP decreased during and following the end of the ice melt period, and was negative from 23–27 April.

One possible explanation for the GOP results is that light may have limited productivity at the beginning of the time-series and nutrients may have limited productivity at the end of the time-series. In this case, the peak in GOP during the end of the melt period could indicate a reduction in light limitation causing enhanced growth rates, followed by the onset of nutrient limitation. Although nutrient concentrations have not been measured in the Bras d'Or Lake during winter or early spring, Strain and Yeats (2002) estimated a maximum wintertime $NO_3^-$ concentration of ~5 $\mu$M, based on fall nutrient measurements and assuming no primary production for 3 months. This $NO_3^-$ concentration should be considered a maximum since the assumption of dormant phytoplankton in winter may not be correct (Hampton et al., 2015). Five $\mu$M of $NO_3^-$ could support gross primary production of 33 mmol C m$^{-3}$, using a Redfield ratio of 106:16 C:N (Redfield, 1963), and 46 mmol $O_2$ m$^{-3}$, using a ratio of 1.4:1 $O_2$:C for nitrate-based uptake (Laws, 1991). This concentration of $NO_3^-$ would be consumed in 8 days at the average volumetric GOP of 5.5 mmol $O_2$ m$^{-3}$ d$^{-1}$, or just 3 days using the average GOP rate from 16–19 April of 15(8) mmol $O_2$ m$^{-3}$ d$^{-1}$.

Another possible explanation for the GOP trends is that nutrients limited productivity throughout the time-series, causing GOP to be similar at the start and end of the time-series. In this case, the likely increase in volumetric GOP during the most rapid reduction in ice cover could be due to changes in the water column that affected nutrient supply (e.g., nutrients were released from the melting ice, or the ice melt drove convection which entrained nutrients from deeper waters into the mixed layer).



The decrease in volumetric NOP must be due to an increase in respiration and recycling of organic carbon by heterotrophs (and possibly also autotrophs) within the mixed layer, since autotrophic production (GOP) stayed relatively constant. One possible explanation for the NOP trends is that zooplankton growth rates increased as the ice was melting and water warmed, leading to an increase in respiration at the same time that GOP was decreasing due to nutrient limitation. Zooplankton growth

rates may increase more rapidly than phytoplankton growth rates in response to increasing temperatures (Rose and Caron, 2007). The water temperature in the system rapidly increased between 12–24 April from ∼2 to 7 °C, as the ice cover retreated and the surface waters absorbed more heat and therefore it is likely that growth rates also increased over this period (Figure 7b).

A second possible explanation for the decreasing NOP is that ice melt (and perhaps riverine inflow) increased the organic

carbon concentrations within the water column and therefore increased grazing rates by heterotrophs. Zooplankton are thought to graze primarily on free-floating phytoplankton and algae that are released from the ice, rather than algae attached to the ice surface, and organic carbon sedimentation events associated with ice melt have been observed in other environments (Michel et al., 2002; Boetius et al., 2013). Concentrations of organic carbon in ice vary widely. Boetius et al. (2013) measured average ice algal deposition to sediments of 750 mmol C m$^{-2}$ during summer in the Arctic. If a similar amount of ice algae were

released from the ice in the Bras d'Or Lake, it would be more than sufficient to support the observed increase in respiration (decrease in NOP) of up to ∼7 mmol O$_2$ m$^{-2}$ d$^{-1}$ (5–6 mmol C m$^{-2}$ d$^{-1}$, using a C:O$_2$ ratio of 1.1–1.4 (Laws, 1991; Bender et al., 1999)) at the end of the time-series. Additionally, river flow on Cape Breton Island typically peaks in April or May, and river flows in April are ∼40-100 % higher than flows in March and ∼50–150 % greater than the average monthly flow. However, dissolved organic carbon concentrations of rivers in Cape Breton during early spring have not been measured and

the bioavailability of terrigenous organic carbon to aquatic microbes is not well known (Wiegner et al., 2006; Caissie and Robichaud, 2009; Guillemette and del Giorgio, 2011).

However, for both of these potential explanations of the NOP results, the increased respiration rates should also increase the nutrient supply available to autotrophs which would counteract nutrient limitation of GOP. Therefore, we might expect the observed increase in respiration (decrease in NOP) to favor an increase in GOP because the organic matter is being more

efficiently recycled and therefore available to support more autotrophic production within the mixed layer. It is possible that the increase in mixed layer depth between 16–23 April diluted the autotrophic biomass and caused volumetric GOP to decrease.

Additionally, the uncertainties in GOP are relatively large, averaging $^{+52}_{-31}$ % for the volumetric GOP throughout the time-series, and therefore it is possible that there were moderate differences in GOP between the beginning and end of the time-series that are not apparent from our methods. For example, a systematic error in the gas transfer velocity at the end of the time-series

(the Injection 2 result) would cause us to systematically overestimate or underestimate GOP at the end of the time series, if our calculated $k_{O_2}$ were too high or too low, respectively. We discuss the potential impacts of other physical processes on our GOP and NOP estimates below (section 3.6).



### 3.5 Comparison to other productivity estimates

To our knowledge, our data are the only published estimates of NOP and GOP in the Bras d'Or Lake. Geen (1965) measured primary production by $^{14}$C uptake in the Bras d'Or Lake during summer 1962–1964 and found an average uptake rate at 5 m depth, the depth of maximum photosynthesis, of ~4 mmol C m$^{-3}$ d$^{-1}$ (50 mg C m$^{-3}$ d$^{-1}$) (Geen and Hargrave, 1966).

These rates are based on 6-h daytime incubations and ignore nighttime respiratory loss of $^{14}$C, and therefore approximate net primary production (NPP, gross primary production minus autotrophic respiration) or something between NPP and gross primary production (GPP) (Bender et al., 1999; Marra, 2002, 2009; Quay et al., 2010). Assuming a C:O$_2$ ratio of 1.1–1.4 (Laws, 1991), the equivalent O$_2$ production based on the $^{14}$C incubations is 4.4–5.6 mmol O$_2$ m$^{-3}$ d$^{-1}$. Thus, the $^{14}$C-PP is between the average NOP and GOP values, as expected.

In the Bras d'Or Lake, Hargrave and Geen (1970) found, based on summertime incubations, that zooplankton grazing was sufficient to consume all of the daily primary production, indicating the estuary ecosystem metabolism may be close to balanced (NOP ~ 0). We obtained a more dynamic record of NOP, with an average volumetric rate of 1.1(2.0) mmol O$_2$ m$^{-3}$ d$^{-1}$ over the entire time-series.

    Comparisons with in situ gas tracer-based productivity estimates in other environments are challenging because mixed layer-

integrated rates are most commonly calculated, and the mixed layer in many other systems is much deeper than 0.8–3 m. In the Beaufort Gyre (Arctic Ocean) Stanley et al. (2015) estimate a steady state GOP of 16(5) and 38(3) mmol O$_2$ m$^{-2}$ d$^{-1}$ in summer 2011 (higher ice cover) and 2012 (lower ice cover), respectively and NOP of 3 mmol O$_2$ m$^{-2}$ d$^{-1}$ in both summers. Mixed layer depths were ~10 m. Mixed layer-integrated GOP increases as the mixed layer depth increases, but trends in NOP are less clear. In this study we found that cumulative volumetric and cumulative mixed layer-integrated NOP were similar when

calculated over the entire time-series, but the result might be different if we integrated to 10 m.

### 3.6 Effect of physical processes on productivity estimates

There are a number of additional environmental processes that may affect the O$_2$ isotope and O$_2$/Ar mass balance in the mixed layer but cannot be directly quantified from the time-series. However, in some cases we can determine whether these processes would tend to increase or decrease our NOP and GOP estimates.

The isotopic composition of freshwater within the bay may have varied during our time-series, which would impact the GOP estimates. Using model results from Gurbutt and Petrie (1995) we calculate that the residence time of water in Whycocomagh Bay is ~0.7 y for surface waters (0–10 m) with respect to freshwater input and ~2 y for deep waters (10 m to bottom) with respect to exchange with the surface waters. Thus we expect the isotopic composition of the water in the mixed layer of Whycocomagh Bay to reflect some average over multiple months. For example, if a substantial portion of the meltwater

entering the estuary is derived from snow rather than from ice that freezes from water within the bay, its isotopic composition will be more reflective of seasonal precipitation. If we calculate an amount-weighted $\delta^{18}$O-H$_2$O for meteoric water at Truro, NS, using only the months when ice was present in Whycocomagh Bay in 2013 (January–April), $\delta^{18}$O-H$_2$O = -11.0(3.6) ‰





versus VSMOW, which is within the uncertainty of our annually-averaged value (-9.3(3.1) ‰). In general, a lower value of $\delta^{18}$O-H$_2$O will increase GOP estimates.

The freezing and melting of ice in saline waters will generate a non-linear salinity-$\delta^{18}$O-H$_2$O relationship because the $\delta^{18}$O-H$_2$O value of sea ice is similar to the water from which it formed (within ∼2–3 ‰) (Tan and Strain, 1980; Macdonald et al.,

1995, 1999), but the salinity is substantially lower due to brine rejection (O'Neil, 1968; Weeks and Ackley, 1986). We are not able to accurately quantify the triple oxygen isotopic composition of H$_2$O in the ice and in water, nor can we quantify the timing and volume of ice freezing and melt within Whycocomagh Bay (although we know when the ice cover decreased most rapidly, the ice volume was decreasing throughout our time-series). The volume contribution and isotopic composition of other sources of water inputs (e.g., riverine input and melting snow) is another source of uncertainty in the calculations.

In addition to affecting the isotopic composition of H$_2$O, ice melt and riverine inflow may affect the NOP and GOP calculations in other ways. If the ice melted at the upper surface (in contact with the atmosphere) and then drained through brine channels in the ice, it likely had an isotopic composition and gas ratio similar to air-equilibrated water ($^{17}\Delta \simeq 8$ per meg and $\Delta$(O$_2$/Ar) $\simeq 0$ %). Thus, water in melt ponds that was added to the water column would tend to decrease GOP, as $^{17}\Delta$ always exceeded 8 per meg in the mixed layer (Figure 6a) and either increase or decrease NOP, since $\Delta$(O$_2$/Ar) transitioned from

negative to positive as the ice was melting. If the ice melted at the bottom (in contact with the water) its effect on NOP and GOP estimates is less clear. During sea ice formation, approximately 40–55 % of the O$_2$ and Ar originally dissolved in the water is retained in the ice matrix (i.e. within the ice itself, in gas bubbles, or in brine pockets) and the remainder is excluded, generating supersaturations of the gases in the water below the ice (Top et al., 1988; Hood et al., 1998; Loose et al., 2009). Photosynthesis and respiration both occur in sea ice (Loose et al., 2011a; Zhou et al., 2014) and will change the O$_2$/Ar and

$^{17}\Delta$ signatures within the ice and it is difficult to predict what proportions of the O$_2$ within brine pockets in the ice remained within the brines, migrated into the water column, or migrated into the atmosphere prior to the complete melting of the ice. We measured an ice thickness of ∼0.3 m near the injection site on 29 March, and thus if the mixed layer depth after ice melt was 2.5 m deep, the ice could contribute ∼11 % of the mixed layer volume, or ∼5 % of the mixed layer O$_2$ (assuming [O$_2$] in ice is ∼45 % of the equilibrium [O$_2$] in water). Thus bottom ice melt would likely be a minor influence on the oxygen mass balance.

We observed bare ice and melt ponds at the surface of the ice (Supplemental Figures 2-4) and the water temperature throughout our time-series was above freezing which would stimulate bottom melt. Therefore, both surface and bottom melt likely occurred during the time-series. The volume, O$_2$ concentration and isotopic composition of runoff and river water during our time-series is also poorly constrained and thus these water sources are another uncertainty in our NOP and GOP calculations.

It is likely that GOP occurred below the mixed layer but was not quantified by our methods because we only had measure-

ments within the mixed layer. In open water, the Secchi depth at the ice edge on 26 March (just west of Little Narrows) was 1.9 m, yielding a euphotic zone depth of ∼5 m (defined as the depth where 1 % of surface photosynthetically active radiation penetrates) and the Secchi depth at Little Narrows on 7 April was ∼4.5 m giving an approximate euphotic zone depth of 12 m (Idso and Gilbert, 1974). Mixed layer depths during our time-series ranged from 0.8–3.0 m. Even when O$_2$ measurements below the euphotic zone are available, it is challenging to quantify this subsurface productivity because the biological O$_2$ fluxes





below the mixed layer are small and the physical fluxes are large and highly uncertain, as they are driven by lateral and vertical mixing rather than air-water gas exchange (Giesbrecht et al., 2012; Munro et al., 2013; Manning et al., 2017b).

We also cannot correct our results for the effect of entrainment of deeper waters into the mixed layer (Hendricks et al., 2004; Munro et al., 2013; Wurgaft et al., 2013). We were not able to sample below the mixed layer but we hypothesize that $^{17}\Delta$ and

$O_2/Ar$ both likely decreased below the mixed layer when the ice was melting and the mixed layer was deepening. Therefore, entrainment of these waters into the mixed layer would tend to decrease NOP and GOP estimates. Respiration increases with depth, causing $[O_2]$ and $\Delta(O_2/Ar)$ to decrease with depth in oceanic systems (Spitzer and Jenkins, 1989; Emerson et al., 1991). In some oceanic regions, photosynthesis below the mixed layer generates excess $^{17}\Delta$ (because there is no process decreasing $^{17}\Delta$ below the mixed layer), which can then be entrained into the euphotic zone, where $^{17}\Delta$ is lower because some of the

photosynthetic $O_2$ is ventilated to the atmosphere (Hendricks et al., 2004; Sarma et al., 2005; Juranek and Quay, 2013). In this study, the gas transfer velocity out of the mixed layer was extremely low up until 16 April, and therefore photosynthetic $O_2$ (i.e., $^{17}\Delta$) would also accumulate in the mixed layer, likely at a greater rate than below the mixed layer due to photosynthesis rates being higher closer to the surface. Once the gas transfer velocity increased to the open water value (20 April), it is not clear whether we would expect to observe an excess of $^{17}\Delta$ below the mixed layer, and therefore whether entrainment would

increase or decrease GOP.

Another source of error in our interpretation of the $O_2$ data is that we must assume that spatial variability in $O_2$ has a negligible effect on our calculations. We interpret all changes in $O_2$ assuming that we are measuring the same water mass; this is an oversimplification because the bay is within an estuary that experiences tidal flows. However, spatial surveys of $O_2/Ar$ during a pilot experiment in 2011 indicated that spatial variability in $O_2$ within the Bras d'Or Lakes is relatively low.

**4   Conclusions**

Using the dual tracer ($^3$He/SF$_6$) technique in the Bras d'Or Lake, we found that at >90 % ice cover, the gas transfer velocity was 6 % of the open water gas transfer velocity. This result indicates that $k_{ice}$ is negligible.

The volumetric GOP was similar at the beginning and end of the time-series, when the basin was full of ice and when it was ice free, but likely increased in the middle of the time-series as the last of the ice melted. Volumetric NOP was more variable

with time; Whycocomagh Bay was on average net autotrophic (NOP > 0) while the ice was melting, and net heterotrophic (NOP < 0) but with a smaller magnitude after the bay was ice-free. These results indicate that an algal bloom (increasing NOP) can occur in an ice-covered estuary, similar to observations in the Great Lakes, Arctic, and Antarctic. The decrease in NOP may be due to the onset of nutrient limitation and/or an increase in heterotrophic respiration of organic carbon released from the ice and possibly also from rivers.

Obtaining a time-series of $O_2$ data and obtaining simultaneous gas transfer velocity measurements were both critical for quantifying productivity. The non-steady state term was a significant contributor to NOP and GOP throughout the time-series, and time-series measurements are needed to quantify the non-steady state $O_2$ flux. Additionally, because the gas transfer velocity was ~16 times higher at the end of the time-series than at the beginning and the values of $^{17}\Delta$ and $\Delta(O_2/Ar)$ changed





with time, the relative importance of the steady state term versus the non-steady state term changed substantially during our time-series.

## 5 Code availability

MATLAB code for calculating GOP incorporating the local isotopic composition of $H_2O$ is available on Zenodo at https:
5 //dx.doi.org/10.5281/zenodo.376786 and GitHub at https://github.com/caramanning/calcGOP (Manning and Howard, 2017).

## 6 Data availability

The discrete $O_2/Ar$ and triple oxygen isotope data, continuous $O_2/Ar$ data, and CTD profile data are available on Zenodo at http://dx.doi.org/10.5281/zenodo.1011216 (Manning and Stanley, 2017).

*Author contributions.* CC Manning, RHR Stanley, B Loose, and B Hatcher designed the study. CC Manning, RHR Stanley, B Loose, B
10 Hatcher, and A Lovely collected the samples. CC Manning, RHR Stanley, B Loose, and P Schlosser analyzed samples and processed data. CC Manning, RHR Stanley, DP Nicholson, and B Loose interpreted the data, and CC Manning prepared the manuscript with contributions from RHR Stanley, DP Nicholson, and B Loose. The authors declare that they have no conflict of interest.

*Acknowledgements.* We thank Zoe Sandwith (WHOI) and Dave Woodland (Cape Breton University) for field and laboratory assistance. We also thank the staff of the Little Narrows Ferry Terminal for providing facilities. RHR Stanley and B Loose received funding from the WHOI
15 Arctic Research Initiative, and CC Manning received graduate scholarships from NSERC and CMOS. Figure 1 (b–c) were modified from an image created by Hanhil (https://commons.wikimedia.org/wiki/File:Canada_Nova_Scotia_location_map_2.svg), "Canada Nova Scotia location map 2" as permitted under the CC-BY-SA 3.0 license, https://creativecommons.org/licenses/by-sa/3.0/legalcode.





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
