# Peer review of "Changes in gross oxygen production, net oxygen production, and air-water gas exchange during seasonal ice melt in Whycocomagh Bay, a Canadian estuary in the Bras d'Or Lake system"

_Biogeosciences, 2017_

## Referee Comment (RC1) · Anonymous Referee #1 · 23 Nov 2017

**Review for manuscript bg-2017-428:**
**"Changes in gross oxygen production, net oxygen production, and air-water gas exchange during seasonal ice melt in the Bras d'Or Lake, a Canadian estuary"** by Manning et al., 2017

In this manuscript the authors present net oxygen and gross oxygen productions (NOP and GOP) in a channel of Whycocomagh Bay, a semi-enclosed estuary in Nova Scotia, Canada.

The authors use the well known $O_2/Ar$ and triple oxygen isotopes technique for determination of aquatic primary production. This method has been widely used in oceanic environments in the past and in this context, the manuscript provides one of the first applications of this method in a brackish water system influenced by ice coverage during winter months.

Few studies to date have done dedicated studies to find out the gas transfer coefficient through ice. This parameter holds the largest uncertainty for the determination of net and gross oxygen productions through the methods applied. In this study by Manning et al., one of the main aims was to measure the gas transfer coefficient through a dual tracer release experiment in the study site, in the presence of and without ice cover. The study is complemented with a time series of $O_2/Ar$ measurements and discrete sampling for triple oxygen isotopes, however all these measurements are only taking place in an eastward channel (Little Narrows) that connects Wycocomagh Bay with the neighbor St. Patricks Channel and the rest of Bras d'Or Lake. This channel was mostly free of ice during their study period. Indeed is in Little Narrows where the exchange of waters in and out the Bay is taking place, however, the authors did not spend some samples to also collect water in the interior of Wycocomagh Bay nor in the vertical water column.

I was however attracted by the addition of the non-steady state term for the determination of NOP and GOP, which often is neglected, but due to, for example, the lack of vertical measurements. Manning et al., 2017 GRL, presented this in the recent past. However, in this manuscript, this term could not be finally fully explained and the conclusions lead to the typical speculations of potential heterotrophy or influence of vertical exchange.

The topic is of relevance and suitable for the journal Biogeosciences. The manuscript is mostly well written, but many typos were found and those are listed in the minor comments below. The figures and tables are clear, but can still be improved by the addition of missing information.

I was also particularly missing information regarding the water structure of Wycocomagh Bay where apparently, anoxic waters can prevail during some parts of the year and those certainly can influence the primary productivity of the estuary. Also, the authors make little effort to explain the horizontal exchange of water through the channel, despite they measure a tidal record through the study period.

Before this manuscript can be published, I encourage the authors to revise thoroughly the manuscript and consider the major and minor comments listed below.

**Major comments:**
**Abstract**
L3 – Here it is misleading, as in the title, to say that the measurements were done in the Bras d'Or Lake, those were actually in Little Narrows, a channel of the Bras d'Or Lake, and tracer measurements in Wycocomagh Bay. A clearer definition of the study site is required: the Bras d'Or Lake system is conformed by a number of estuaries and channels and also the Bras d'Or Lake itself. It is therefore misleading to say that the measurements where done IN the Bras d'Or Lake. This should be better written and even the title should be modified to reflect the real study site.

**Introduction.**
Throughout the introduction I am missing more information regarding Whycocomagh Bay (WB) for example: how relevant it is, based on previous literature reports, the lateral exchange of water through Little Narrows? How this impacts the biogeochemistry of WB? Is there a well-stratified water column there with little deep-water ventilation? Tides seem to be more important that what the authors suggest initially, more on this point is suggested in the following comment.
I am particularly surprised that the authors do not make any reference to previous works where a detailed description of the physical and biological characteristics in the waters of WB is presented. This is for example the case of Lambert, 2002 (see full reference at the end of this review) where the author presents a detailed description on the characteristics of the different components of the Bras d'Or Lake system. In that work is particularly mentioned that the waters of WB might become low oxygenated in some areas due to restricted circulation that is regulated by topography and coastal configuration. Also in that study, the author suggests that sewage might be an important source of nutrients to the waters of west WB, causing periods of eutrophication, hence, low oxygen conditions. These background information must be added to the introduction.

**Method**
I am surprised that the authors did not make efforts to also sample for $O_2/Ar$ and triple oxygen isotope measurements for waters under the ice in the interior of Wycocomagh Bay, or at least, before the release of tracer in the borehole they had to build. Also, it is well known that this method requires at best also the knowledge of the vertical distribution of $O_2/Ar$ and $\delta^{17}O$ and $\delta^{18}O$, and the authors could also sample in the vertical water column for examination. I think the authors should sustain why this was not done.

P4, L22-24 - This sentence needs more explanation and evidence. From previous published works, how are the tides affect the exchange of waters in WB and influence its biology, oxygen content and physical water column structure?. A published work by Dupont et al., 2003, discusses thoroughly the role of tides in the Bras d'Or Lakes system.
In pag. 10 L27, the authors mention that "visually" the tides seem to play an important role. If ADCP measurements were done, will be good to add some more analysis on this direction and try to find out the effect on the time change

in tides to the observations in $O_2/Ar$, for example. Please also explain Figure 1 of the supplementary material.

Section 2.2 - Can the authors spend some lines at the end of this section, explaining why the concentration of the tracers between injections is different, and also the bubbling period? Why during injection 1 in the presence of ice the tracer was more diluted? Due to expected less ventilation of the tracer?

P17, L10-11 – The non-steady state term of Eq. 9, assumes that there are changes over time of $^{17}\Delta$, and those are related to changes in $h$ due to potential entrainment into or export from the mixed layer. These are physical processes that are only discussed later. The authors should spend some lines here to introduce these explanations here, as to which processes would influence the non-steady state term.

– Besides of addressing my comment of why the selection of the first period to calculate $r^2$ in the $^{17}\Delta$ results until 19 April (mentioned below), will be good to plot a correlation between mixed layer depth and $^{17}\Delta$ to figure out if there is some influence on the $^{17}\Delta$ due to shoaling or deepening events.

**Specific comments:**
**Suggestions of text modifications or additions are written in bold text**
P1, abstract
- Consider introducing the acronym NOP and GOP already in the abstract
- Here and throughout the manuscript, the convention to write the mean value and the standard deviation as: "mean(std. dev.)" should be avoided, and use the traditional "mean±std. dev." format.
P2,
L9 – Add a space between "waters" and "(Ducklow et al."
L10 - "**The** parameterization of gas exchange…"
L11-12 – repetitive sentence to same page L15-16, re-arrange so it does not sound repetitive.
L20-25 – These lines are misleading, by reading them, it gives the impression that the work was done in the Bras d'Or Lake itself. It is until line 25 when reference to Whycocomagh Bay (WB) was made. I would try to move these lines to a new section about "site description", it could be a first section of the methods part or even a sub-section in the introduction.  I would start this paragraph by introducing first WB (L25-29) and briefly mention where it is located, complementing with more background information (previous works) on the physical and biogeochemical properties of the water of WB (mentioned on major comments above).
L22 – "…and has a **total** surface area of …"
L29-30 – Introduce and rephrase: "**The annual** maximum ice cover is typically reached in early March. Ice disappears rapidly during April until its total melt by the first week of May"
L31-33 – These lines should have less detail, and leave it to the methods section, it is enough if it says: "To determine the gas transfer coefficient during the period of study, we performed a two dual-tracer release experiment " or something similar.

P4
L1 – "The **NOP/GOP** ratio, provides …"
L2 – " … production available for export **to waters below** the mixed layer."
L5 - I think here it is fair to still refer to Luz et al., 1999 and Luz and Barkan, 2000 as the initiators of the triple oxygen isotopes method for ocean productivity, rather than the review of Juranek and Quay, 2013 only. The last can still be referenced because it includes also the $O_2/Ar$ ratio method for NOP.
L12 - " (Figure 1) during **a 33-day time series** (25 March-28 April 2013)".
It also is important to mention here that all the measurements were done at the water within the surface mixed layer or simply in waters below the surface.
L33 – "…from a thermocouple **located** in the **sampling** bucket, because it had …"
P5
L2 – " … to correct for warming **effects**."
L6 – How far in cm or m was located the CT sensor from the cable ferry?
L16 – Freshwater inputs to the bay from melting ice and snow, precipitation?
L20 – As I read the manuscript, I am looking for the information on sea-ice thickness in the site of study during the period of analysis; I only found this information in Pag. 21 L11, will be good to move this information to this line for example, ice thickness encountered during Injection 1.
L21 – " … **were** diluted by a factor of …"
L23 – how far was the "separate hole" from the injection site?
P6
L6 – flask evacuation meaning when the flasks were prepared (evacuated) in the laboratory previous to the study?
L12 – which value the authors obtained for $^{17}\Delta$ in equilibrated waters?
L14 – "… 20 mL **of** water in 50 mL glass …"
L22 – how did the authors detected when the tracer was flowing with this system?
L31 – "The Lott and Jenkins**' solubility value** is ~2 % higher…"
P7
L9 – " … **fully covered** by ice, and completed …"
L15 – Here the ice volume and thickness during the sampling period should also be provided.
L19 - what was the vertical resolution of the CTD profiles? Is the data in Figure 3 the result of some vertical interpolation to the CTD measurements?
L20 – how the authors determined the depth of the mixed layer (also dashed line in Figure 3)? Here it should be stated.
L21 – " … the length of the CTD **instrument** and obtaining … "
P9
L11 – Add here what $h$ in Eq. 2 means, and NOT until P10, L17.
P10
L25 – " … m d-1, **in the presence of ice and shallow mixed layers, with the standard deviation of the distribution of $k_{600}$ from the Monte Carlo simulation as the uncertainty value** (Figure 4)."
L26-27 – here refer to Figure 1 of supplementary material and interpret these observations based on the ADCP data if possible, rather than referring to visual observations. Waters flowing into WB can be seen in the eastward velocities panel?. Also, Figure 4 should appear before Table 1, or refer to table 1 earlier in the manuscript.

L29 – "…, and it is possible **that** the…"
L30 – why the authors assume here that the mixed layer was deeper than 0.8±0.3 m during 9-10 April?
P11
L6 – " m d$^{-1}$ **for open waters** (Figure 4)."
L8 – "… lower **in the presence of ice** and 0.4 % lower **in open waters**."
P12
L5-14 - these lines should be moved to the introduction section, e.g., to pag. 2 and after L19.
L16 – "During **visual** surveys along the shoreline of Whycocomagh Bay …"
P13
L5 – what is the surface area of WB? You only gave the total surface area of Bras d'Or Lake in P2, L22. Maybe add this info on WB also in that same paragraph in the intro.
L7 – "… for both injections **($k$=0.71)**, then the results yield …"
L20 – Here it is better to write "time-series until the end of 15$^{th}$ of April" isn't? It would be better to write it like this if the linear interpolation to obtain $k$ in the period where there were no measurements was done from 16 April (from 00:00 h) until end of 20 April.
L22 – "midnight 20 April" means the start of 21 April? Then I would write instead "21 April" to keep consistency with the periods above.
L25 – So which $k$ value was used for the period between 16 and 19 April? Would it make sense to use an average $k$ value between that period of time?. This might be OK since for the period of injection 1 and injection 2, one single $k$ value was used in each case. Please complete here what approach was taken.
L31 – "(Figure 1**b**)"
L32 – "**After** injection 2, between 20 April … "
L33 – "… 2.2 m s-1. The calculated $k_{600}$ over this time period … "
P14
L11 – "… short-term **wind speed** measurements…"
L12 – "… the gas transfer than wind speed **values** averaged over …"
L20 – "… with other studies showing **that** gas transfer velocity … "
L24 – remove "polynyas". Here it only applies "ice leads"
L25 – "**ice leads** that are common in Arctic and Antarctic **sea ice**".
L26 – here I am also missing more information on the ice thickness found during the study. Elaborate further on the statement of "differences in gas transfer behavior are expected based on the nature of the ice pack"
L31 – "dissolved O$_2$ derived from air-water exchange" and remove the text "(from the atmosphere)".
L32- Remove the text "(from H$_2$O)". Later you explain with detail these differences but here it is not explicit enough and just confusing.
P15
L5 – "O$_2$, and **for** H$_2$O is referenced to the …"
L7 – "… quantifies the **$^{17}$O-excess** in dissolved O$_2$."
L13 – 294 per meg should be 249 per meg
L25 – a period is missing between "salinity" and "A similar … "
L30 – remove units for salinity
L31 – " … meteoric **water** endmember … "
L32 – " … study area, **and** 40 m elevation) … "

P16

L5 - What is actually the mean salinity in WB? Please give also this information and not only for Little Narrows, I suspect there is a strong seasonal gradient, but how far from Little Narrows this is actually taking place?

L11 – I don't actually understand why you call $^{17}\Delta$ and $^{17}$O-excess for the dissolved oxygen with atmospheric signal and the one from the water. $^{17}\Delta$ is the $^{17}$O-excess. If you want to make this difference I would then call it $^{17}\Delta_{H2O}$ or $^{17}\Delta_w$ as you did for the slope value, however is still think there is no need to have two separate equations for that, you can simply explain each term what would they imply for case of the atm. $O_2$ and $O_2$ from $H_2O$ but only using a general equation.

P17

L6- and what about $[O_2]_{eq}$?

L9 – the steady state GOP term is also the equation of Prokopenko et al., 2011, please cite her here after the comma.

L15 – How much was the $^{17}\Delta_{eq}$ value for equilibrated water at < 5 °C?

L20 - Is not until 28 April?

L20 – How many samples were collected in average/day?

L22 – refer here to Eq. 9 after "GOP term."

L28-29 – Why the chosen period to end was 19 of April instead of 16 of April to keep consistency with the *k* estimates? I see that from the ML- and volumetric-GOP (Figure 6 c and d), would make sense to connect all the first period until 19 of April, but could have also be done until 16 April and this can be clearly seen in the $^{17}\Delta$ values (Figure 6 a and b). Did the authors test that combination? Actually by looking later to the NOP results, the NOP starts to decrease from 16 April, so it would make sense to consider the first period of GOP also until before 16 April.

P20

L3 – instead of "intermediate ice cover" write "varying ice cover"

L8 – "… 19 April), **because is the time period when the ice retreated rapidly and there was more light availability**, despite the increasing gas transfer velocity."

P21

L6 - "… especially **under thin** first-year ice…"

L11 – this information on ice thickness becomes too late in the manuscript, move it to Pag. 5.

P22

L19 – title of section 3.4.2 has twice "of"

L26 – instead of "decreased" use "retreated"

P24

L3 – "… appears to decrease during …"

L3 – as mentioned earlier, the changes in NOP appear to be contrasting when the transition of sea-ice melting period starts in 16 April, I would still strongly suggest to modify the GOP first period also to 16 April instead to until 19 April, so during that transition NOP decreases and GOP increases, and the only way that this can take place is if at the same time that GOP increases also respiration increases (net heterotrophy) as discussed below. Also the entrainment of deeper waters as the mixed layer deepens, contributes to changes in GOP during the transition period.

L6-7 – what about the entrainment of low oxygenated waters as the mixed layer deepens?

L13-14 – the fact that there were no measurements below the mixed layer does not imply that vertical mixing is not an important process and should still be discussed.

L16 – In Castro-Morales et al., 2013, the influence of entrainment and vertical mixing in mixed layer GOP and NOP are thoroughly discussed. This citation should be included here.

L34 – what about nutrients sources from land (eutrophication), and also through Little Narrows during tidal exchange? How about a correlation between ice coverage and tides? Horizontal exchange in this region seems more relevant, warmer water entering WB influencing ice melt.

P25

L23-26 – Residence time calculations might help supporting this statement ($t=h/k$), please add this information considering the changes in $k$ and $h$ through the time series.

L26 – I agree in this potential explanation, considering that there might be low oxygenated water below the mixed layer in this region.

P26

L12 – " $\sim$ 0) **on a daily basis**."

P27

L5 – "… but the salinity **in sea ice** is substantially lower … "

P28

L4 & 10 – add also to the list of references Castro-Morales et al., 2013

L29 – "… also from rivers, **or due to the entrainment of poor oxygenated waters**."

**Tables**
- Include a small table (or embed this information in Table 1) where includes the following data: full experiment period with dates from start of the experiment (25 March) until the end (28 April), also when the injections took place, and not only the dates for determination of $k$. Include in the table also the ice coverage/fraction. It is hard now to follow all these changes between beginning of tracers experiments, beginning of the full experiment, evolution of ice coverage, etc.

**Figures**
**Figure 1 –** In panel b, add latitude and longitude values in axis, as in panel a. In panel c, add some labels, e.g., Greenland, Canada. The colors in this panel are too subtle and hard to see.

**Figure 3 –** Why the uppermost measurement did not start from the same depth ? I understand that maybe in the presence of ice this was not possible, but what about in open water?

Is there also a dashed vertical line in the last day of sampling (28 April)? Was a last CTD cast conducted that day?

At best, salinity "units" should be removed everywhere.

Wasn't 25 of March the first day of sampling? Why there is data from even earlier than 24 March? Refer in the caption to Table 1 for specific dates of the vertical dashed lines.

**Figure 4 –** There is a typo in the legend for Injection 2, it should be 0.71 and not 0.071. In the line corresponding to Injection 1, the second to fourth measurements point (between 8th and 10th day) it looks actually an increasing tracers ratio. How can the authors explain this? In the end, the line for injection 1 was drawn by three points, initial, day 9 and day 11 (or end), would this still be valid?

**Figure 6 –** In the caption, it should be written at the end " … GOP term are most uncertain **due to melting of sea-ice cover**."

**Figure 7** – Would be good to also add in panels a and b the mixed layer depth using a secondary y-axis to contrast the time evolution of temperature and $O_2/Ar$ with MLD. In the caption it should be changed "yellow diamonds and pink triangles" to "green diamonds and blue squares" isn't?

**References**
- P1, L16 - Here Loose and Schlosser, 2011 should be "2011b". Loose et al., 2011b should become 2011c.
- Add the doi's to all the references where it is missing
- in Eveleth et al., 2016, change to full citation (not in press anymore), vol. 139, 89-102, 10.1016/j.dsr2.2016.07.016.

**References listed in this review**

Castro-Morales, K., N. Cassar, D. R. Shoosmith and J. Kaiser. Biological production in the Bellingshausen Sea from oxygen-to-argon ratios and oxygen tripe isotopes, 2013. Biogeosciences, Vol. 10, pag. 2273-2291.

Dupont, F., B. Petrie, J. Chaffey, 2003. Modeling the tides of the Bras d'Or Lakes, in: Canadian Technical Report of Hydrography and Ocean Sciences 230. Ocean Sciences Division, Maritimes Region, Department of Fisheries and Oceans, Bedford Institute of Oceanography, Dartmouth, Nova Scotia, Canada. (http://www.bio-iob.gc.ca/science/research-recherche/ocean/webtide/documents/model_brasdor.pdf)

Lambert, T., 2002. Overview of the Ecology of the Bras d'Or Lakes with emphasis on the fish. Proc. N.S. Inst. Sci. Vol. 42, Part 1, pag. 65-99.

Luz B., Barkan E., Bender M.L., Thiemens M.H., Boering K.A. 1999. Triple-isotope composition of atmospheric oxygen as a tracer of biosphere productivity. Nature, 400, pag. 547–50.

Luz B., Barkan E. 2000. Assessment of oceanic productivity with the triple-isotope composition of dissolved oxygen. Science, 288, pag. 2028–31.

---

## Referee Comment (RC2) · Anonymous Referee #2 · 24 Jan 2018

Review of Manning et al **Changes in gross oxygen production, net oxygen production, and air-water gas exchange during seasonal ice melt in the Bras d'Or Lake, a Canadian estuary"**

The study of Manning et al presents 1) estimates of gas transfer velocity under almost complete ice cover and ice-free waters; and 2) measurements of Net and Gross Oxygen production rates under ice, during and immediately after the spring ice melt, in Whycocomagh Bay, a semienclosed estuary in Nova Scotia. The study described in the manuscript is novel and has important implications for studies of the polar regions. I am very enthusiastic about the first part of the paper, focused on the gas exchange, but have more reservations about several aspects of the 2$^{nd}$ part of the paper (estimates of NOP and GOP). Thus, I have just a few comments on the former, and outline more detailed concerned on the latter. Overall, I recommend the paper for publication after major revisions.

**I. Gas exchange estimates**
Estimating rates of gas exchange in ice-covered seas is of great importance for biogeochemical studies in polar and subpolar regions, yet gas piston velocities have been notoriously difficult terms to estimate, and remained poorly constrained. The authors used a clever approach, where they released a dual tracer (3He/SF6) and monitored the change in the two tracers ratio through time. All physical processes (mixing, dilution within the water column) except for gas exchange with atmosphere affected both tracers equally, while the difference in solubilities of the two gases controlled the changes in the tracer ratio thus allowing for robust determination of piston velocities. I believe this part of the study is of a substantial scientific value and is a great contribution to future studies of polar regions.

Specific comments

1) The ice edge on April 7$^{th}$ was located at approx. 1 km from the sampling site (based on Fig. 1a). I wonder what was the influence of this open water patch on the estimated k600? From Figure 4, it looks like the 3He/SF6 ratio dropped substantially on Day 10. What would be k value if this last point is excluded from the fit?
2) It would be useful to give the actual values of molecular diffisitivities for 3He and SF6 (e.g. somewhere in Section 3.2.1)
3) Is there a reason for the 3He/SF6 increase in the first 3 data points after Injection 1? Or this is likely a "noise" signal? This issue is addressed on p. 10, line 23-25, but some clarification would good. What was the tidal status when these 3 data points were taken (other than "visual observations" mentioned in the subsequent lines on the same page)?

Typos:
There is a typo in Fig. 4 (k600 of the 2$^{nd}$ injection should be 0.7, not 0.07)
P. 6, line 30 should read " The Lott and Jenkins solubility is ~2% higher"

**II. Estimates of NOP and GOP rates**
The 2$^{nd}$ part of the study was devoted to estimates of NOP and GOP at the Little Narrows sampling location. The productivity terms have been poorly constrained in ice covered and partially ice covered high latitude seas. Thus, the value of this part of the study is in expanding our (currently very limited) knowledge of these terms in the polar regions, despite some limitations (in setup, calculations and interpretation), which I address next.

Specific comments

1. Setup of the study:
O2/Ar monitored and samples for $^{17}\Delta$ were collected at the Little Narrows, which has been ice free all through the length of the time series (based on Figure 1a). How valid it is to apply the piston velocity determined for the ice covered conditions at the ice free Little Narrow study site?
To address this issue, a more through description of the Little Narrows study site is needed: what is residence time of surface water in this channel? Current velocities? In other words, how well measured here O2/Ar and $^{17}\Delta$ signals represent the conditions within the Bay? While the authors do state that spatial variability within the Bay is likely to be small, the issue at hand here is – are the reported NOP and GOP rates really the rates under ice (for the time period between March 31 and April 18$^{th}$) or are they more representative of the very local ice free waters in the Little Narrows?

2. Calculations:
Equation (5) modified from Prokopenko et al 2011 (equation S8 in that paper, would be good to give a citation) contains two terms, O2 and h (in the NSS term). In Prokopenko et al, O2 and h were assumed constant, while the $^{17}\Delta$ was time-dependent. However, in the study of Manning et al, this is clearly not the case, particularly for the O2 term. In fact, there appears to be a discrepancy between Equation (5) and (11). In the former, O2 is treated as if it in a steady state, while in Equation 11 O2 (as O2/Ar) is treated as a time-variable term.

So, one question is what were the actual values of O2 that the authors used to calculate the reported GOP terms for every time point? Similarly, what were the values of h (though the latter is probably less important, at least for the period between March 30-April 10, when mixed layer depth remained more or less constant, however the changes in h after April 10$^{th}$ are more substantial)?

Based on the above, I am not sure that the approach chosen by the authors to calculate the NSS term is fully correct. I would suggest that the authors redo the calculations using the approach presented in Haskell et al (2017), published in Global Biogeochemical Cycles, which presents an alternative treatment of the NSS term in NOP and GOP calculations under non-steady state conditions.

3. Interpretation:
An obvious problem of the study is the lack of information about the water column below the mixed layer. Are there any published studies on the water column winter conditions?

Clearly, TOI was presented by Manning is the fist of this kind in the Bay. But it would be very important to know the degree of oxygenation of the water column during winter months. If the Bay goes anoxic (or very low O2), estimates of NOP are impossible to make without knowing how much of this winter low O2 signal contributes to the mixed layer.

The problem is likely less acute for GOP, as the absolute difference between the water column and the mixed layer $^{17}\Delta$ is likely to be smaller than for O2/Ar (and one could assume $^{17}\Delta$ being as low as at atmospheric equilibrium value). However, the drop in GOP observed after the ice melt is really difficult to explain: obviously, increased ventilation should not lower GOP, thus the drop of $^{17}\Delta$ is driven, in addition to increased ventilation, by dilution of the mixed layer $^{17}\Delta$ with waters carrying lower 17O-excess. GOP does not require new nutrients (as the authors themselves point out), thus even when all the nutrients are consumed, GOP should not be affected.

The problem with NOP estimates is likely be more acute, but since the O2 concentration below the mixed layer is not known, it is more difficult to assess quantitatively.

On the positive side, this study provides a very interesting and novel example of using the 17O-excess method in waters different from VSMOW, and this part of work is of great value.

Smaller comments:
P. 15, line 22 - Pls, add original reference for "published relationship between d17O-H2O and d18O-H2O , where only the reference to Manning et al (2017) is currently given).
P. 17, line 15-17 – what was the actual values for the fractionation factor used in calculations (as based on Luz and Barkan, 2011)?
P. 20, line 20 should read " the mixed layer-integrate GOP"

In summary, my recommendation for Interpretation part would be to re-write the discussion of the NOP and GOP estimates, clearly stating the above limitations first and substantially shorten the discussion of zooplankton and heterotrophy. It appears that the estimates of NOP rates are likely strongly affected by mixing of the O2-depleted signal and it would not be possible to evaluate the magnitude of this effect). As to GOP – I wonder if using the approach of Haskell would modify the estimated GOP trend through the time series (and some estimates of the degree of dilution of $^{17}\Delta$ signal could be made).

---

## Author Comment (AC1) · 6 Jan 2019

**Re: Biogeosciences manuscripb bg-2017-428**
**Author Comment in response to Review 2**

Dear Editor and Reviewers:

We would like to thank Reviewer 2 for their thorough review of the manuscript.  Reviewer 2 felt the paper had substantial scientific value and recommended publication in Biogeosciences. They were enthusiastic about the gas exchange aspect of the study and recommended major revisions to the productivity portion of the paper before publication.

Please find our detailed response to Reviewer 2 below.  Reviewer comments are in black and our responses are in blue.

Thank you,
Cara Manning
* * *
Review of Manning et al Changes in gross oxygen production, net oxygen production, and air-water gas exchange during seasonal ice melt in the Bras d'Or
Lake, a Canadian estuary"

The study of Manning et al presents 1) estimates of gas transfer velocity under almost complete ice cover and ice-free waters; and 2) measurements of Net and Gross Oxygen production rates under ice, during and immediately after the spring ice melt, in Whycocomagh Bay, a semienclosed estuary in Nova Scotia. The study described in the manuscript is novel and has important implications for studies of the polar regions. I am very enthusiastic about the first part of the paper, focused on the gas exchange, but have more reservations about several aspects of the 2nd part of the paper (estimates of NOP and GOP). Thus, I have just a few comments on the former, and outline more detailed concerned on the latter. Overall, I recommend the paper for publication after major revisions.

I. Gas exchange estimates
Estimating rates of gas exchange in ice-covered seas is of great importance for biogeochemical studies in polar and subpolar regions, yet gas piston velocities have been notoriously difficult terms to estimate, and remained poorly constrained. The authors used a clever approach, where they released a dual tracer (3He/SF6) and monitored the change in the two tracers ratio through time. All physical processes (mixing, dilution within the water column) except for gas exchange with atmosphere affected both tracers equally, while the difference in solubilities of the two gases controlled the changes in the tracer ratio thus allowing for robust determination of piston velocities. I believe this part of the study is of a substantial scientific value and is a great contribution to future studies

of polar regions.

Thank you for these positive comments.

Specific comments
1) The ice edge on April 7th was located at approx. 1 km from the sampling site (based on Fig. 1a). I wonder what was the influence of this open water patch on the estimated k600? From Figure 4, it looks like the 3He/SF6 ratio dropped substantially on Day 10. What would be k value if this last point is excluded from the fit?

The position of the ice edge was already included in the interpretation of the Injection 1 results (new/modified text is in italics):
"We mapped out the location of the ice edge closest to Little Narrows by boat on 26 March, 7 April, and 12 April (Figure 2a). Using these surveys and shoreline data, we calculate that for the surface area of the bay between the injection site and Little Narrows, f = 0.01 on 26 March, 0.05 on 7 April, and 0.08 on 12 April. The f experienced by the tracer patch during Injection 1 is likely between *0.02*--0.08 because the tracer *was injected on 30--31 Mar* and flowed through the open water near Little Narrows between 6--11 April."

We also added in the following text:
*"One source of uncertainty in estimating the correct value of f is that the transit speed between the injection site and Little Narrows was non-constant. The mean current velocity at Little Narrows channel was 3.4 km d$^{-1}$, but the tracer took approximately 8 days to flow 7 km from the injection site to Little Narrows."*

Regarding the selection of data points, we added in the following text (in italics):
"The tracer ratio did not display a consistent decrease over the three days we sampled it at Little Narrows, *which may be* due to the substantially lower gas transfer velocity, *as well as the very low tracer concentrations potentially increasing measurement uncertainty*."

We performed a sensitivity analysis when calculating *k* for each injection. For Injection 1, if the day 10 measurement is excluded from the fit, $k_{600}$ is 0.0485 (0.0024) m d$^{-1}$, 7% higher than the value reported above. Including (excluding) the measurement on day 10, $k_{600}$ for injection 1 is 6% (7%) of the value for injection 2. Given the small difference in *k* when day 10 is added or removed, we think it is most appropriate to include all 4 measurements.

2) It would be useful to give the actual values of molecular diffisitivities for 3He and SF6 (e.g. somewhere in Section 3.2.1)
We added in the Schmidt numbers of the two gases in this section (the ratio of diffusivity to kinematic viscosity of water): *"The Schmidt number at a salinity of 4 PSS and temperature of 2 $^o$C is 305 for $^3$He and 2684 for SF$_6$."*

3) Is there a reason for the 3He/SF6 increase in the first 3 data points after Injection 1? Or this is likely a "noise" signal? This issue is addressed on p. 10, line 23-25, but some clarification would good. What was the tidal status when these 3 data points were taken (other than "visual observations" mentioned in the subsequent lines on the same page)?

This is likely a noise effect.  We stated on page 10, line 23;
"For Injection 1, the excess SF6  and 3 He concentrations were reduced by two orders of magnitude by the time the tracer reached Little Narrows (7–11 days after injection). The tracer ratio did not display a consistent decrease over the three days we sampled it at Little Narrows, likely in part due to the substantially lower gas transfer velocity."

The concentration of both tracers was lower than expected by the time the tracer reached Little Narrows (as the transit time was longer than expected), so there is some uncertainty in the ratio in each individual measurement. However, the calculated $k$ is relatively insensitive to our choice of data points.

Typos:
There is a typo in Fig. 4 (k600 of the 2nd injection should be 0.7, not 0.07)

Than you for noticing this important error. We have fixed the figure as shown below:

[Figure]

P. 6, line 30 should read " The Lott and Jenkins solubility is ~2% higher"
Done.

II. Estimates of NOP and GOP rates
The 2nd part of the study was devoted to estimates of NOP and GOP at the Little Narrows

sampling location. The productivity terms have been poorly constrained in ice covered and partially ice covered high latitude seas. Thus, the value of this part of the study is in expanding our (currently very limited) knowledge of these terms in the polar regions, despite some limitations (in setup, calculations and interpretation), which I address next.

Specific comments

1. Setup of the study:

O2/Ar monitored and samples for $17\Delta$ were collected at the Little Narrows, which has been ice free all through the length of the time series (based on Figure 1a). How valid it is to apply the piston velocity determined for the ice covered conditions at the ice free Little Narrow study site?

To address this issue, a more through description of the Little Narrows study site is needed: what is residence time of surface water in this channel? Current velocities? In other words, how well measured here O2/Ar and $17\Delta$ signals represent the conditions within the Bay? While the authors do state that spatial variability within the Bay is likely to be small, the issue at hand here is – are the reported NOP and GOP rates really the rates under ice (for the time period between March 31 and April 18th) or are they more representative of the very local ice free waters in the Little Narrows?

Regarding current velocity, we added in the following text in the section "sampling setup at Little Narrows":

*"We deployed a Nortek acoustic Doppler current profiler (ADCP) at ~4 m depth in the middle of the channel from 7--28 Apr. The mean current speed at 0.5 m depth was 3.4 km $d^{-1}$ (3.9 cm $s^{-1}$) with an orientation toward $31^o$ (approximately along the channel axis), indicating the transit time through Little Narrows is relatively short."*

Regarding the influence of the open water between the ice edge and Little Narrows on the GOP/NOP calculations, we added in the following text in the section on GOP results:

*"We believe that the $O_2$ measurements at Little Narrows from the beginning of the time-series through 16 April (the date when the fraction of open water in Whycocomagh Bay began to increase rapidly) are generally representative of the under-ice rates due to the rapid transport of water through Little Narrows channel. The mean current speed at Little Narrows channel at 0.5 m depth was 3.4 km $d^{-1}$. The distance between the ice edge and the sampling intake ranged from ~0.4–1 km between the beginning of the time-series and 12 Apr. Assuming a gas transfer velocity for $O_2$ of 0.5 m $d^{-1}$ once the water mass encountered open water and a mixed layer depth between 0.8 to 2 m, the residence time of $O_2$ in the mixed layer was 1.6–4 d. Assuming a current speed of 3.4 km in open water, the water mass would have only been exposed to open water for between 0.1–0.3 d (20 % or less of the residence time of $O_2$). However, we recognize that the current speed near the ice edge may have been somewhat slower than the speed at the constriction at Little Narrows, therefore the transit time in open water may have been somewhat longer than 0.1–0.3 d."*

2. Calculations:

Equation (5) modified from Prokopenko et al 2011 (equation S8 in that paper, would be

good to give a citation) contains two terms, O2 and h (in the NSS term). In Prokopenko et al, O2 and h were assumed constant, while the $_{17}\Delta$ was time-dependent. However, in the study of Manning et al, this is clearly not the case, particularly for the O2 term. In fact, there appears to be a discrepancy between Equation (5) and (11). In the former, O2 is treated as if it in a steady state, while in Equation 11 O2 (as O2/Ar) is treated as a time-variable term.

So, one question is what were the actual values of O2 that the authors used to calculate the reported GOP terms for every time point? Similarly, what were the values of h (though the latter is probably less important, at least for the period between March 30-April 10, when mixed layer depth remained more or less constant, however the changes in h after April 10$_{th}$ are more substantial)?

Based on the above, I am not sure that the approach chosen by the authors to calculate the NSS term is fully correct. I would suggest that the authors redo the calculations using the approach presented in Haskell et al (2017), published in Global Biogeochemical Cycles, which presents an alternative treatment of the NSS term in NOP and GOP calculations under non-steady state conditions.

Thank you for these suggestions. We think you were referring above to equation (9) of our paper, which is the equation for GOP, rather than equation (5), which is the equation for $k_{eff}$. We have added the citation to Prokopenko et al. (2011) as you suggested.

As indicated in equation (10), the value of $[O_2]$ used to calculate GOP was determined from the $O_2/Ar$ ratio of the sample and assuming $[Ar]$ is at equilibrium, i.e.
$$[O_2] = \Delta(O_2/Ar)_{samp} [O_2]_{eq}.$$

The value of $h$ (mixed layer depth) was the instantaneous value interpolated from our CTD profiles (black dot-dashed line in Figure 3 a and b).

You state above that "in Prokopenko et al., $O_2$ and h were assumed constant, while $^{17}\Delta$ was time dependent." We believe that is incorrect. In Prokopenko et al. (2011), equation 5, which is time-dependent in $O_2$ concentration and isotopic composition, is used to derive equation S8. In other words, the time-dependency in $O_2$ concentration is already built into eq. 5 and S8. It is correct that the change in mixed layer depth with time is not included. However, we have concluded that including a time-variable mixed layer depth in the GOP and NOP calculations would require knowledge of the gradient in $O_2$ concentration and isotopic composition as a function of depth below the mixed layer. For example, when $O_2$ diffuses upward into the mixed layer, the rate of $O_2$ addition to the mixed layer will depend on the $O_2$ concentration below the mixed layer. This must be added as an additional term in the $O_2$ mass balance equation. For example, see Howard et al. (2010) equation 9, Nicholson et al. (2014) equations 12 and 13 and Munro et al. (2013) equation 5 and 7.

Additionally, we reached out to the authors of Haskell et al. (2017). It is true that equation 1a in Haskell et al. (2017) incorporates a non-steady state mixed layer, whereas the equation we used did not. However, when we carefully derived an equation with a non-steady state mixed layer

depth, we did not achieve the same equation as in Haskell et al. (2017). In particular, we believe that equation 1a Haskell et al. (2017) double counts the change in $O_2$ by including it within the time rate of change term because equations 5 and S8 in Prokopenko et al. (2011) already account for the time rate of change in $O_2$ concentration without it appearing directly in the equation. Thus, we are concerned that there may be a mistake in equation 1a of Haskell et al. (2017) and do not want to use it for this paper.

The reviewer stated that including a time-dependent $O_2$ term is very important and that including a time-dependent mixed layer term is less important. Since we have already done the former and are concerned about the accuracy of the Haskell et al. (2017) equation, we have decided to leave the GOP calculations unchanged in the revised manuscript.

3. Interpretation:
An obvious problem of the study is the lack of information about the water column below the mixed layer. Are there any published studies on the water column winter conditions? Clearly, TOI was presented by Manning is the first of this kind in the Bay. But it would be very important to know the degree of oxygenation of the water column during winter months. If the Bay goes anoxic (or very low O2), estimates of NOP are impossible to make without knowing how much of this winter low O2 signal contributes to the mixed layer.

We now state in the section on the sampling setup:
*"This CTD package was also equipped with an $O_2$ sensor, but unfortunately the $O_2$ sensor malfunctioned throughout the experiment. Therefore, we can characterize vertical structure of salinity and temperature but not $O_2$."*
We did not mention this detail in the original manuscript in the interest of brevity but have added it to better explain our experimental design.

Regarding subsurface $O_2$ levels, we now added the following text in the section "site description", a new section that Reviewer 1 recommended we include:
*"To our knowledge, there are no previously published measurements of water column chemistry in Whycocomagh Bay in ice-covered conditions. Measurements from July 1974 showed that the western portion of the Whycocomagh Bay became anoxic in the isolated deep waters below 25 m depth whereas measurements in the eastern portion of the basin (closer to Little Narrows) showed the water column had an $O_2$ saturation of 61% at the bottom depth of 30 m (Krauel, 1975; Gurbutt and Petrie, 1995). Measurements collected from 1995–1997 (from late April to late September) showed $O_2$ concentrations in Whycocomagh Bay from 1–5 m depth were near equilibrium (94 to 112 % saturation) throughout the bay. In the deeper waters, $O_2$ concentrations in eastern Whycocomagh Bay ranged from 69 % at 25 m depth on 28 April 1996 to 54 % at 13 m depth on 26 Sep 1995 and 30 % at 30 m depth on 23 Sep 1996, and the western basin was persistently anoxic (Strain et al., 2001)."*

We regret not obtaining a Niskin bottle for subsurface discrete sample measurements of triple oxygen isotopes and have added this recommendation for future work to our conclusions section.

*"In future studies in similar settings, we recommend authors collect measurements of $O_2/Ar$ and triple oxygen isotopes below the mixed layer in order to better correct for the impact of vertical processes on the $O_2$ mass balance. For tidally-influenced regions, surveys of lateral variability in $O_2/Ar$ and $^{17}\Delta$ could also help to constrain the influence of horizontal advection on the $O_2$ mass balance (Munro et al. 2013, Howard et al. 2017)."*

The problem is likely less acute for GOP, as the absolute difference between the water column and the mixed layer $17\Delta$ is likely to be smaller than for $O_2/Ar$ (and one could assume $17\Delta$ being as low as at atmospheric equilibrium value). However, the drop in GOP observed after the ice melt is really difficult to explain: obviously, increased ventilation should not lower GOP, thus the drop of $17\Delta$ is driven, in addition to increased ventilation, by dilution of the mixed layer $17\Delta$ with waters carrying lower 17O-excess. GOP does not require new nutrients (as the authors themselves point out), thus even when all the nutrients are consumed, GOP should not be affected.

The problem with NOP estimates is likely be more acute, but since the O2 concentration below the mixed layer is not known, it is more difficult to assess quantitatively.

As discussed in our response to Reviewer 1, we have now eliminated the GOP and NOP estimates during the period 16-19 April where the estimates are very uncertain due to uncertainty in the parameterization of *k*. Additionally, we have included more text emphasizing the uncertainties associated with the variable mixed layer depth and lack of depth-resolved measurements. We eliminated much of the discussion of explanations for the decreasing NOP at the end of the time-series.

New text added to the section on GOP "results and discussion":
*"Before interpreting the GOP results, we emphasize that the impact of changing mixed layer depth on the $O_2$ mass balance is not accounted for in our calculations. The mixed layer deepened from 8--23 Apr and then appeared to shoal from 23--28 Apr. If the $^{17}\Delta$ value below the mixed layer were higher (lower) than $^{17}\Delta$ in the mixed layer, this would cause us to overestimate (underestimate) GOP as the mixed layer deepened."*

New text added at the beginning of section "NOP results and comparison of NOP and GOP":
*"As for GOP, we do not present NOP estimates during 16–19 Apr due to the uncertainties associated with the rapid decrease in ice cover. There are additional uncertainties in NOP from 12–23 Apr due to the increasing mixed layer depth, which may have entrained lower-$O_2$ waters into the mixed layer. Historical $O_2$ data from Whycocomagh Bay showed that there is a strong decrease in $O_2$ with depth below the mixed layer, however, to our knowledge, no prior measurements during ice melt have been published (Krauel, 1975; Gurbutt and Petrie, 1995; Strain et al., 2001)."*

New text added at the end of section "NOP results and comparison of NOP and GOP"
*"The mixed layer depth was ~0.8 m from the beginning of the time-series until 12 April, deepened from 12–23 Apr and then increased until the end of the time-series. The apparent decrease in volumetric NOP beginning around 12 April could be due to a number of factors*

*including: an increase in respiration and recycling of organic carbon by autotrophs and/or heterotrophs, and/or vertical processes that are not taken into account in our calculations. We note that the mixed layer NOP was actually lowest and ΔO$_2$/Ar was decreasing between 23–27 Apr, a period where the mixed layer actually appeared to shoal. Therefore a deepening mixed layer cannot be the only driver of the decrease in NOP following ice melt."*

On the positive side, this study provides a very interesting and novel example of using the 17O-excess method in waters different from VSMOW, and this part of work is of great value.
Thank you.

Smaller comments:
P. 15, line 22 - Pls, add original reference for "published relationship between d17O-H2O and d18O-H2O , where only the reference to Manning et al (2017) is currently given).
We've added citations to Luz et al (2010) and Li et al (2016).

P. 17, line 15-17 – what was the actual values for the fractionation factor used in calculations (as based on Luz and Barkan, 2011)?
We now specify "$\alpha^{18}_p$ = 1.0033890 and $\alpha^{17}_p$ = 1.0017781."

P. 20, line 20 should read " the mixed layer-integrate GOP"
Done.

In summary, my recommendation for Interpretation part would be to re-write the discussion of the NOP and GOP estimates, clearly stating the above limitations first and substantially shorten the discussion of zooplankton and heterotrophy. It appears that the estimates of NOP rates are likely strongly affected by mixing of the O2-depleted signal and it would not be possible to evaluate the magnitude of this effect). As to GOP – I wonder if using the approach of Haskell would modify the estimated GOP trend through the time series (and some estimates of the degree of dilution of $_{17}\Delta$ signal could be made).

As discussed above, we have continued to use the equation of Prokopenko et al. (2011) to calculate GOP, as this equation already incorporates non-steady state O$_2$ concentration and isotopic compostion. We believe that correcting the GOP estimates for non-steady state mixed layer would require knowledge of the O$_2$ concentration below the mixed layer, data which is not available to us. We have highlighted the uncertainties as you suggested.

Thank you again for your thorough review and suggestions that have improved our manuscript.

---

## Author Comment (AC2) · 6 Jan 2019

**Re: Biogeosciences manuscript bg-2017-428**
**Author Comment in response to Review 1**

Dear Editor and Reviewers:

We would like to thank Reviewer 1 for their careful and constructive review of the manuscript. Reviewer 1 commented that the paper is novel and appropriate for Biogeosciences because it is one of the first applications of the triple oxygen isotope method in brackish waters influenced by ice cover, and that few studies have determined the gas transfer coefficient through ice. They had some concerns about the lack of correction for vertical mixing in the productivity calculations and made various additional major and minor suggestions that they requested we address before the manuscript is published.

Please find our detailed response to Reviewer 1 below. Reviewer comments are in black and our responses are in blue.

Thank you,
Cara Manning
* * *
Review for manuscript bg-2017-428:
"Changes in gross oxygen production, net oxygen production, and air-water gas exchange during seasonal ice melt in the Bras d'Or Lake, a Canadian estuary" by Manning et al., 2017

In this manuscript the authors present net oxygen and gross oxygen productions (NOP and GOP) in a channel of Whycocomagh Bay, a semi-enclosed estuary in Nova Scotia, Canada.

The authors use the well known $O_2/Ar$ and triple oxygen isotopes technique for determination of aquatic primary production. This method has been widely used in oceanic environments in the past and in this context, the manuscript provides one of the first applications of this method in a brackish water system influenced by ice coverage during winter months.

Few studies to date have done dedicated studies to find out the gas transfer coefficient through ice. This parameter holds the largest uncertainty for the determination of net and gross oxygen productions through the methods applied. In this study by Manning et al., one of the main aims was to measure the gas transfer coefficient through a dual tracer release experiment in the study site, in the presence of and without ice cover.

The study is complemented with a time series of O2/Ar measurements and discrete sampling for triple oxygen isotopes, however all these measurements are only taking place in an eastward channel (Little Narrows) that connects Wycocomagh Bay with the neighbor St. Patricks Channel and the rest of Bras d'Or Lake. This channel was mostly free of ice during their study period. Indeed is in Little Narrows where the exchange of waters in and out the Bay is taking place,

however, the authors did not spend some samples to also collect water in the interior of Wycocomagh Bay nor in the vertical water column.

I was however attracted by the addition of the non-steady state term for the determination of NOP and GOP, which often is neglected, but due to, for example, the lack of vertical measurements. Manning et al., 2017 GRL, presented this in the recent past. However, in this manuscript, this term could not be finally fully explained and the conclusions lead to the typical speculations of potential heterotrophy or influence of vertical exchange.

The topic is of relevance and suitable for the journal Biogeosciences. The manuscript is mostly well written, but many typos were found and those are listed in the minor comments below. The figures and tables are clear, but can still be improved by the addition of missing information.

Thank you for these comments and for taking the time to list the typos you noticed. We have corrected these typos and added additional information to the figures and tables as detailed below.

I was also particularly missing information regarding the water structure of Wycocomagh Bay where apparently, anoxic waters can prevail during some parts of the year and those certainly can influence the primary productivity of the estuary. Also, the authors make little effort to explain the horizontal exchange of water through the channel, despite they measure a tidal record through the study period.

We added a new section "site description" after the introduction, as you suggested later in this review. This section includes discussion of historical data on water column structure (temperature, salinity, oxygen) and currents. Please see our responses to your detailed comments below.

For example, regarding tidal exchange, we added the following to the "site description" section: "*In general, excluding the narrow channels in direct contact with the ocean, the Bras d'Or Lakes system does not exhibit significant tidally-driven variability in temperature, salinity, and sea surface level (Krauel, 1975; Petrie, 1999; Petrie and Bugden, 2002).*"

Regarding historical $O_2$ data, we added in the following:
*"To our knowledge, there are no previously published measurements of water column chemistry in Whycocomagh Bay in ice-covered conditions. Measurements from July 1974 showed that the western portion of the Whycocomagh Bay became anoxic in the isolated deep waters below 25 m depth whereas measurements in the eastern portion of the basin (closer to Little Narrows) showed the water column had an $O_2$ saturation of 61% at the bottom depth of 30 m (Krauel, 1975; Gurbutt and Petrie, 1995). Measurements collected from 1995–1997 (from late April to late September) showed $O_2$ concentrations in Whycocomagh Bay from 1–5 m depth were near equilibrium (94 to 112 % saturation) throughout the bay. In the deeper waters, $O_2$ concentrations in eastern Whycocomagh Bay ranged from 69 % at 25 m depth on 28 April 1996 to 54 % at 13 m depth on 26 Sep 1995 and 30 % at 30 m depth on 23 Sep 1996, and the western basin was persistently anoxic (Strain et al., 2001).*"

Before this manuscript can be published, I encourage the authors to revise thoroughly the manuscript and consider the major and minor comments listed below.

Major comments:
Abstract
L3 – Here it is misleading, as in the title, to say that the measurements were done in the Bras d'Or Lake, those were actually in Little Narrows, a channel of the Bras d'Or Lake, and tracer measurements in Wycocomagh Bay. A clearer definition of the study site is required: the Bras d'Or Lake system is conformed by a number of estuaries and channels and also the Bras d'Or Lake itself. It is therefore misleading to say that the measurements where done IN the Bras d'Or Lake. This should be better written and even the title should be modified to reflect the real study site.

We have changed the article title to: "Changes in gross oxygen production, net oxygen production, and air-water gas exchange during seasonal ice melt in Whycocomagh Bay, *a Canadian estuary in the Bras d'Or Lake system.*"

Thanks for correcting our misunderstanding about the difference between the Bras d'Or Lake and the Bras d'Or Lake system.

Reference: http://www4.rncan.gc.ca/search-place-names/unique/CAEYO

Introduction.
Throughout the introduction I am missing more information regarding Whycocomagh Bay (WB) for example: how relevant it is, based on previous literature reports, the lateral exchange of water through Little Narrows? How this impacts the biogeochemistry of WB? Is there a well-stratified water column there with little deep-water ventilation? Tides seem to be more important that what the authors suggest initially, more on this point is suggested in the following comment.

We added in a new section "site description" which appears after the introduction. We added the following text in regard to lateral exchange of water through Little Narrows.

*"The deep water in western Whycocomagh Bay is strongly isolated due to the presence of the sill, Little Narrows Channel, and very little mixing occurs between the surface and sub-surface waters (Petrie and Bugden, 2002). Whycocomagh Bay receives a relatively large amount of freshwater compared to other regions of the bay. This freshwater forms ice in winter and the stratification remains very stable, preventing vertical mixing (Krauel, 1975). In general, excluding the narrow channels in direct contact with the ocean, the Bras d'Or Lakes system does not exhibit significant tidally-driven variability in temperature, salinity, and sea surface level (Krauel, 1975; Petrie, 1999; Petrie and Bugden, 2002)."*

I am particularly surprised that the authors do not make any reference to previous works where a detailed description of the physical and biological characteristics in the waters of WB is presented. This is for example the case of Lambert, 2002 (see full reference at the end of this review) where the author presents a detailed description on the characteristics of the different components of the Bras d'Or Lake system. In that work is particularly mentioned that the waters of WB might become low oxygenated in some areas due to restricted circulation that is regulated by topography and coastal configuration.  Also in that study, the author suggests that sewage might be an important source of nutrients to the waters of west WB, causing periods of eutrophication, hence, low oxygen conditions. These background information must be added to the introduction.

We had cited other papers on the Bras d'Or Lake system that appeared in the same conference proceedings as the Lambert paper mentioned above (e.g. Petrie and Bugden 2002) but have now included additional references with background information on the Bras d'Or Lake system, particularly with regards to subsurface $O_2$ and currents (e.g. Krauel, 1975; Petrie, 1999; Petrie et al, 2002; Dupont et al. 2003) in the new section "site description."

Method
I am surprised that the authors did not make efforts to also sample for O2/Ar and triple oxygen isotope measurements for waters under the ice in the interior of Wycocomagh Bay, or at least, before the release of tracer in the borehole they had to build. Also, it is well known that this method requires at best also the knowledge of the vertical distribution of O2/Ar and δ17O and δ18O, and the authors could also sample in the vertical water column for examination. I think the authors should sustain why this was not done.

These are some details that we originally left out in the interest of brevity, as the manuscript was already quite long, but we have now included them.

We have added the following text (in italics) to the manuscript in the section "sampling setup at Little Narrows":
**"We collected conductivity, temperature and depth (CTD) profiles with a SBE 19Plus sensor at Little Narrows, usually by boat using a winch, but occasionally by lowering the CTD by hand on a rope from the Little Narrows cable ferry. *This CTD package was also equipped with an $O_2$ sensor, but unfortunately the $O_2$ sensor malfunctioned throughout the experiment. Therefore, we can characterize vertical structure of salinity and temperature but not $O_2$."***

We have added the following text to the manuscript in the section "tracer injections":
*"We also collected under-ice samples for $O_2$/Ar and TOI immediately before and after the injection but they displayed a wide range in values for $O_2$/Ar, from -14 to 2%. Therefore, we were unable to define any initial under-ice values for these parameters."*

To provide a bit more detail on the latter point: we collected $O_2$/Ar and triple oxygen isotope measurements at the site of Injection 1 and these data are included in the dataset we have published in association with this paper. Two samples collected under the ice before the injection started had a $\Delta O_2$/Ar of 2% with excellent reproducibility, whereas six samples collected after

the injection had $\Delta O_2/Ar$ ranging from -14 to -9%. Given this wide range of values, we decided it would be inappropriate to report any "initial" values for $O_2/Ar$ and TOI based on this under ice data. The significant mixing that occurred as this water mass transited from the injection site to Little Narrows would also complicate the use of these initial samples to calculate a productivity rate.

When this study was planned, in 2012 and early 2013, many of the papers describing quantitative methods for correcting TOI data for vertical fluxes had not yet been published. We now recognize that vertical profiles of TOI would have greatly improved the study. We have added the following text to the section "conclusions":
*"In future studies in similar settings, we recommend authors collect measurements of $O_2/Ar$ and triple oxygen isotopes below the mixed layer in order to better correct for the impact of vertical processes on the $O_2$ mass balance. For tidally-influenced regions, surveys of lateral variability in $O_2/Ar$ and $^{17}\Delta$ could also help to constrain the influence of horizontal advection on the $O_2$ mass balance (Munro et al. 2013, Howard et al. 2017)."*

P4, L22-24 - This sentence needs more explanation and evidence. From previous published works, how are the tides affect the exchange of waters in WB and influence its biology, oxygen content and physical water column structure?. A published work by Dupont et al., 2003, discusses thoroughly the role of tides in the Bras d'Or Lakes system.
In pag. 10 L27, the authors mention that "visually" the tides seem to play an important role. If ADCP measurements were done, will be good to add some more analysis on this direction and try to find out the effect on the time change in tides to the observations in O2/Ar, for example. Please also explain Figure 1 of the supplementary material.

We added the following text (in italics) at P4, L22-24: "*The mean current speed at 0.5 m depth was 3.4 km $d^{-1}$ (3.9 cm $s^{-1}$) with an orientation toward $31^o$ (approximately along the channel axis), indicating the transit time through Little Narrows is relatively short.* The ADCP data is shown in Supplemental Figure 1. Our measurements (salinity, $O_2$/Ar, etc.) did not display any correlation with tidal cycles, *consistent with previous studies of the Bras d'Or Lakes system indicating tides have a negligible influence on water properties within Whycocomagh Bay (Krauel, 1975; Petrie, 1999; Petrie and Bugden, 2002; Dupont et al., 2003)."*

Section 2.2 - Can the authors spend some lines at the end of this section, explaining why the concentration of the tracers between injections is different, and also the bubbling period? Why during injection 1 in the presence of ice the tracer was more diluted? Due to expected less ventilation of the tracer?

We added in text to section 2.2 to explain the concentration of tracers.

Regarding Injection 1: *"Because the tracer was added at a fixed location, the tracer was bubbled very slowly to increase the fraction of the gas that dissolved."*

Regarding Injection 2: *"We used a higher quantity and concentration of $SF_6$ and $^3He$ during this injection because we expected the tracer would be ventilated more rapidly due to higher gas exchange rates given the open water conditions. We were able to inject the tracer more rapidly because it was distributed over a large area instead of through a small hole in the ice."*

P17, L10-11 – The non-steady state term of Eq. 9, assumes that there are changes over time of $17\Delta$, and those are related to changes in h due to potential entrainment into or export from the mixed layer. These are physical processes that are only discussed later. The authors should spend some lines here to introduce these explanations here, as to which processes would influence the non-steady state term.

We have added the following text to section 3.3.1 on the calculation of GOP in regards to your and Reviewer 2's suggestions.

*"This non-steady state equation for GOP incorporates temporal variability in both $O_2$ concentration and isotopic composition (see equation 5 in Prokopenko et al. (2011)). It does not account for time-variability in mixed layer depth (e.g. entrainment of deeper waters into the mixed layer), nor does it account for vertical diffusion of $O_2$ across the mixed layer. In order to account for vertical processes in the $O_2$ mass balance, we would need to have triple oxygen isotope measurements below the mixed layer (Castro Morales et al., 2013; Munro et al., 2013; Wurgaft et al., 2013; Nicholson et al., 2014). Since we do not have these data, we cannot correct for vertical processes affecting the $O_2$ budget."*

Please note that equation 9 is derived from an equation that assumes that the rate of change in the inventory of each $O_2$ isotopologue with time is proportionate to the rate of O2 production, minus the rate of respiratory $O_2$ loss, minus the loss $O_2$ due to gas exchange. It does not incorporate changing mixed layer depth.

Besides of addressing my comment of why the selection of the first period to calculate r2 in the $17\Delta$ results until 19 April (mentioned below), will be good to plot a correlation between mixed layer depth and $17\Delta$ to figure out if there is some influence on the $17\Delta$ due to shoaling or deepening events.

In response to comments from both reviewers, we decided to no longer calculate GOP or NOP during the ice melt period because of the uncertainties in the steady-state and non-steady state term at this time. In Figures 6 and 7 we have replaced the subplot of cumulative NOP and GOP (panel e) with the mixed layer depth so that the relationship between $^{17}\Delta$ and mixed layer depth can be seen more clearly.

Specific comments:
Suggestions of text modifications or additions are written in bold text
P1, abstract
- Consider introducing the acronym NOP and GOP already in the abstract

Done.

- Here and throughout the manuscript, the convention to write the mean value and the standard deviation as: "mean(std. dev.)" should be avoided, and use the traditional "mean±std. dev." format.
We have removed the parentheses from the error reporting throughout the manuscript.

P2,
L9 – Add a space between "waters" and "(Ducklow et al."
Done.

L10 - "**The** parameterization of gas exchange…"
Done.

L11-12 – repetitive sentence to same page L15-16, re-arrange so it does not sound repetitive.
This paragraph has been rewritten as follows:
*"Many investigators have assumed that there is negligible gas transfer through ice and therefore the gas transfer velocity can be linearly scaled as a function of the fraction of open water, multiplied by the open water gas transfer velocity (Takahashi et al. 2009, Legge et al., 2015, Evans et al. 2015, Stanley et al., 2015). A recent field study by Butterworth and Miller (2016) in the Southern Ocean at 0-100% ice cover verified this approach. However, other studies report that gas exchange is reduced or enhanced in the presence of sea ice relative to a linear scaling based on the fraction of open water, including some studies measuring higher transfer velocities in ice-covered waters than in open water (Fanning and Torres, 1991; Else et al., 2011; Papakyriakou and Miller, 2011; Rutgers van der Loeff et al., 2014)."*

L20-25 – These lines are misleading, by reading them, it gives the impression that the work was done in the Bras d'Or Lake itself. It is until line 25 when reference to Whycocomagh Bay (WB) was made. I would try to move these lines to a new section about "site description", it could be a first section of the methods part or even a sub-section in the introduction. I would start this paragraph by introducing first WB (L25-29) and briefly mention where it is located, complementing with more background information (previous works) on the physical and biogeochemical properties of the water of WB (mentioned on major comments above).

In this section, we now refer to the study site as *"Whycocomagh Bay, an estuary in the Bras d'Or Lake system on Cape Breton Island in Nova Scotia, Canada."* As you suggested, and as we mentioned above, we have added a subsection titled "site description" after the introduction where more references with background information are now cited.

L22 – "…and has a **total** surface area of …"
Done.

L29-30 – Introduce and rephrase: "**The annual** maximum ice cover is typically reached in early March. Ice disappears rapidly during April until its total melt by the first week of May"
Done.

L31-33 – These lines should have less detail, and leave it to the methods section, it is enough if it says: "To determine the gas transfer coefficient during the period of study, we performed a two dual-tracer release experiment " or something similar.
Rewritten as *"We performed two dual tracer release experiments at different fractions of ice cover to quantify air-water gas exchange by adding $^3He$ and $SF_6$ to the mixed layer."*

P4
L1 – "The **NOP/GOP** ratio, provides …"
Done.

L2 – " … production available for export **to waters below** the mixed layer."
Done.

L5 - I think here it is fair to still refer to Luz et al., 1999 and Luz and Barkan, 2000 as the initiators of the triple oxygen isotopes method for ocean productivity, rather than the review of Juranek and Quay, 2013 only. The last can still be referenced because it includes also the O2/Ar ratio method for NOP.
We have added the citations from 1999 and 2000 as suggested.

L12 - " (Figure 1) **during a 33-day time series** (25 March-28 April 2013)".
It also is important to mention here that all the measurements were done at the water within the surface mixed layer or simply in waters below the surface.
Changed as followed (new text in bold)
"We continuously sampled water at Little Narrows channel (Figure 1) **during a 33-day time series** (25 March--28 April 2013). We moored a Goulds SB Bruiser 5-18 GPM submersible pump with intake at ~0.5 m depth **(within the mixed layer)**,"

L33 – "…from a thermocouple **located** in the **sampling** bucket, because it had …"
Done.

P5
L2 – " ... to correct for warming **effects**."
Done.

L6 – How far in cm or m was located the CT sensor from the cable ferry?
The distance varied depending on where the cable ferry was located. We do not have the exact distance but it was on the order of at least 100 m.

L16 – Freshwater inputs to the bay from melting ice and snow, precipitation?
We have added the text in italics: "freshwater inputs to the bay *from surface runoff and precipitation."*

L20 – As I read the manuscript, I am looking for the information on sea-ice thickness in the site of study during the period of analysis; I only found this information in Pag. 21 L11, will be good to move this information to this line for example, ice thickness encountered during Injection 1. We added in that *"The ice thickness was ~0.3 m and the injection occurred in the upper 0.5 m of the water column."* Unfortunately we only measured the ice thickness at the injection site and one other location, both of which were relatively close to shore and not near the ice edge in Whycocomagh Bay. We did not do a full survey of ice thickness across the bay due to the safety hazards associated with working on thinning first-year ice.

L21 – " … **were** diluted by a factor of …"
Done.

L23 – how far was the "separate hole" from the injection site?
We have added the following text in this section: "We sampled for initial $^3$He and $SF_6$ concentrations at the injection site, immediately before starting the injection, and after terminating the tracer addition, *by drilling several holes >10 m away from the injection site (because the bubbling action would have perturbed gas concentrations at the injection site).*"

P6
L6 – flask evacuation meaning when the flasks were prepared (evacuated) in the laboratory previous to the study?
Yes. The reason for reporting this is because there can be very slow leakage of air across the o-rings of the flasks.

L12 – which value the authors obtained for 17Δ in equilibrated waters?
The value was 8.1 per meg. This was given on page 17, line 14, in the section on GOP calculation methods. We think it is more appropriate for the air-equilibrated water value to be reported on page 17.

L14 – "… 20 mL **of** water in 50 mL glass …"
Done.

L22 – how did the authors detected when the tracer was flowing with this system?
We have rephrased this section as:
*"When the automated system showed measurable $SF_6$ in the water, we began collecting discrete samples for $SF_6$ and $^3$He."*

L31 – "The Lott and Jenkins' **solubility value** is ~2 % higher…"
Reworded as: The Lott and Jenkins *solubility values are ~2%* higher.

P7
L9 – " … **fully covered** by ice, and completed …"

Done.

L15 – Here the ice volume and thickness during the sampling period should also be provided.
We previously stated "Changes in surface ice cover and total ice volume are both important factors during the study; changes in ice volume/thickness will affect stratification and convection in the mixed layer as well as light penetration through the ice, and the surface ice cover affects the rate of gas exchange "

This was stated to alert the reader that there could be changes in the biogeochemistry before the ice cover had fully retreated.

Ice thickness measurements are reported later in the manuscript (section "tracer injections"). Unfortunately we do not have sufficient data to report the ice volume. We feel it would be inaccurate to extrapolate an ice volume based on measurements at 2 locations at only one time.

L19 - what was the vertical resolution of the CTD profiles? Is the data in Figure 3 the result of some vertical interpolation to the CTD measurements?
We now state in the Figure 3 caption:  "The data in this plot were binned into 0.2 m depth bins from 0.4 to 3.0 m and 0.3 m depth bins from 3.3 to 9.9 m.  The shallowest depth varies between casts due to challenges in getting stable CTD data in the upper 1 m."

Near-surface data where it was clear that the CTD had not stabilized (which had unrealistically high temperature and/or low salinity values) was eliminated.

L20 – how the authors determined the depth of the mixed layer (also dashed line in Figure 3)? Here it should be stated.
This is already explained in the subsequent paragraph, line 23-29.

L21 – " … the length of the CTD instrument and obtaining … "
Done.

P9
L11 – Add here what h in Eq. 2 means, and NOT until P10, L17.
Thanks, we've added this.

P10
L25 – " … m d-1**, in the presence of ice and shallow mixed layers, with the standard deviation of the distribution of k600 from the Monte Carlo simulation as the uncertainty value** (Figure 4)."
Done.

L26-27 – here refer to Figure 1 of supplementary material and interpret these observations based on the ADCP data if possible, rather than referring to visual observations. Waters flowing into WB can be seen in the eastward velocities panel?. Also, Figure 4 should appear before Table 1, or refer to table 1 earlier in the manuscript.

Figure 4 now appears before Table 1.

Yes, because it is tidally influenced, there is net transport into Whycocomagh Bay during some periods. However, we found that the mean velocity over our measurement period was 3.4 km/d with an orientation of 31$^{o}$ (northeast, approximately along channel).

L29 – "…, and it is possible **that** the…"
Done.

L30 – why the authors assume here that the mixed layer was deeper than 0.8±0.3 m during 9-10 April?
We have added a reference to Figure 3 (which shows the mixed layer increased some time between April 8 and 12).

P11
L6 – " m d-1 **for open waters** (Figure 4)."
Done.

L8 – "… lower **in the presence of ice** and 0.4 % lower **in open waters**."
Done.

P12
L5-14 - these lines should be moved to the introduction section, e.g., to pag. 2 and after L19.
We have moved L9-14 to page 2, after line 19.
*"An additional factor reducing air-water exchange in ice-covered waters is that ice significantly reduces fetch for wave generation and therefore wind-driven near-surface turbulence (Squire et al., 1995; Overeem et al., 2011). However, other processes may enhance near-surface turbulence in the presence of sea ice including convection associated with heat loss and brine rejection (Morison et al., 1992; Smith and Morison, 1993), boundary layer shear between ice and water (McPhee, 1992; Saucier et al., 2004), and wave interactions with drifting ice (Kohout and Meylan, 2008)."*

We did not move L5-8 to page 2 because these lines reference equation 5 and $k_{ice}$. This equation and term is not presented until page 12 and we prefer to keep the introduction qualitative.

L16 – "During visual surveys along the shoreline of Whycocomagh Bay …"
Done.

P13
L5 – what is the surface area of WB? You only gave the total surface area of Bras d'Or Lake in
P2, L22. Maybe add this info on WB also in that same paragraph in the intro.
We don't know the exact surface area of WB, however, we already reported on page 2 that
Whycocomagh Bay is "approximately 13 km long and 3 km wide" and provide scale bars on
Figures 1a and 2.

L7 – "… for both injections **(k=0.71)**, then the results yield …"
Changed to "k = 0.71 m d$^{-1}$."

L20 – Here it is better to write "time-series until the end of 15th of April" isn't? It would be
better to write it like this if the linear interpolation to obtain k in the period where there were no
measurements was done from 16 April (from 00:00 h until end of 20 April).
Rewritten as suggested.

L22 – "midnight 20 April" means the start of 21 April? Then I would write instead "21 April" to
keep consistency with the periods above.
Done.

L25 – So which k value was used for the period between 16 and 19 April? Would it make sense
to use an average k value between that period of time?. This might be OK since for the period of
injection 1 and injection 2, one single k value was used in each case. Please complete here what
approach was taken.
This was stated on Line 24-25: "Between 16 April and 20 April, we apply a linear interpolation
of the $k_{600}$ for Injection 1 and Injection 2 as a function of time." The estimated k is shown in
Figure 6f but we do not cite that figure here because we think it makes more sense for the figure
to appear in the manuscript later, when all of the GOP calculations are reported.

L31 – "(Figure 1**b**)"
Done.

L32 – "**After** injection 2, between 20 April … "
Reworded as: "During the period that samples with tracer from Injection 2 were collected
(between 20 April 23:00 and 23 April 11:00)".

L33 – "… 2.2 m s-1. The calculated k600 over this time period … "
Done.

P14
L11 – "… short-term **wind speed** measurements…"
Done.

L12 – "… the gas transfer than wind speed **values** averaged over …"
Done.

L20 – "… with other studies showing **that** gas transfer velocity … "
Done.

L24 – remove "polynyas". Here it only applies "ice leads"
Done.

L25 – "**ice leads** that are common in Arctic and Antarctic **sea ice**".
Done.

L26 – here I am also missing more information on the ice thickness found during the study. Elaborate further on the statement of "differences in gas transfer behavior are expected based on the nature of the ice pack"
The ice thickness is now reported. We have limited measurements of ice thickness due to safety considerations.

L31 – "dissolved O2 derived from air-water exchange" and remove the text "(from the atmosphere)".
Done.

L32- Remove the text "(from H2O)". Later you explain with detail these differences but here it is not explicit enough and just confusing.
Done.

P15
L5 – "O2, and **for** H2O is referenced to the …"
Changed to: "isotopes of $H_2O$ are referenced to the..."

L7 – "… quantifies the 17O-excess in dissolved O2."
We have not changed this. We prefer to not use $^{17}$O-excess to refer only to the isotopic composition of $H_2O$ and not to $O_2$. Please see our response to your comment on P15, L 11.

L13 – 294 per meg should be 249 per meg
Changed to 242 per meg and cited Manning et al. (2017).

L25 – a period is missing between "salinity" and "A similar … "
Done.

L30 – remove units for salinity
We have retained PSS when referring to salinity values. PSS is an acronym for Practical Salinity Scale and is not a unit. We feel that specifying PSS is necessary. There are multiple salinity scales commonly used (ppt, g/kg, PSS, etc.), especially in estuarine science, and without this acronym it is not clear which scale was used.

Reference: Millero (1993) What is PSU? https://tos.org/oceanography/assets/docs/6-3_letter.pdf

L31 – " … meteoric **water** endmember … "
Done.

L32 – " … study area**, and** 40 m elevation) … "
Done.

P16
L5 -  What is actually the mean salinity in WB? Please give also this information and not only for Little Narrows, I suspect there is a strong seasonal gradient, but how far from Little Narrows this is actually taking place?
We expect near-surface salinity throughout Whycocomagh Bay to be lowest during ice melt. Unfortunately, as there are no previously published under-ice measurements of salinity in Whycocomagh Bay, we cannot comment on the mean salinity in Whycocomagh Bay or how it evolves seasonally. There is also very little published data at 1 m depth or shallower, which would be the most relevant depth for comparison to our study, as our pump intake was at 0.5 m depth and the mixed layer was very shallow.

L11 – I don't actually understand why you call 17$\Delta$ and 17O-excess for the dissolved oxygen with atmospheric signal and the one from the water. 17$\Delta$ is the 17O-excess. If you want to make this difference I would then call it 17$\Delta$H2O or 17$\Delta$w as you did for the slope value, however is still think there is no need to have two separate equations for that, you can simply explain each term what would they imply for case of the atm. O2 and O2 from H2O but only using a general equation.
We have added in the following sentence:
*"Equations 7 and 8 have a similar form, however, researchers in the $H_2O$ isotope community have traditionally used the $^{17}O$-excess terminology, whereas researchers in the $O_2$ isotope community have used the $^{17}\Delta$ notation (Luz & Barkan, 2010)."*

P17
L6- and what about [O2]eq?
The difference between $[O_2]$ and $[O_2]_{eq}$ was described on line 9-10.

L9 – the steady state GOP term is also the equation of Prokopenko et al., 2011, please cite her here after the comma.

We added her citation before line 6: " We calculate GOP using equation 7 from Manning et al. (2017), which is equivalent to equation S8 from Prokopenko et al. (2011)"

L15 – How much was the $^{17}\Delta$eq value for equilibrated water at < 5 °C?

This was not measured, probably due to temperature limitations of the equilibration apparatus. However, because there was no significant difference between values at 20 and 5 $^{o}$C, we expect the value at 0 $^{o}$C (the lowest temperature observed in our study) would be similar. Many researchers at various institutions have measured the $^{17}\Delta$ of air-equilibrated water as a function of temperature and found that, after applying appropriate size corrections to the raw mass spectrometer data, there is no measurable effect of temperature on $^{17}\Delta_{eq}$. These size corrections are necessary because a fresh water sample equilibrated at 5 $^{o}$C will contain 40% more $O_2$ than a sample equilibrated at 20 $^{o}$C.

L20 - Is not until 28 April?

There is no oxygen isotope data available for 28 April (the only sample collected was lost during analysis). However, there was continuous $O_2/Ar$ data available through 28 April.

L20 – How many samples were collected in average/day?

We have added that the mean number of samples per day was 2.

L22 – refer here to Eq. 9 after "GOP term."

We have not added in this reference here because all of the GOP calculations use equation 9, referencing it here would suggest it was only used for the non-steady state GOP term.

L28-29 – Why the chosen period to end was 19 of April instead of 16 of April to keep consistency with the k estimates? I see that from the ML- and volumetric-GOP (Figure 6 c and d), would make sense to connect all the first period until 19 of April, but could have also be done until 16 April and this can be clearly seen in the $^{17}\Delta$ values (Figure 6 a and b). Did the authors test that combination? Actually by looking later to the NOP results, the NOP starts to decrease from 16 April, so it would make sense to consider the first period of GOP also until before 16 April.

Lines 30-35 indicate why we chose to have the time period for the first regression end on 19 April. We tried a number of different options for the regression.

Based on comments from Reviewer 1 and 2 we decided to eliminate GOP and NOP estimates during the period from 16-19 April where the rate of change in $^{17}\Delta$ with time, the gas transfer velocity, and the influence of changing mixed layer are all large sources of uncertainty.

P20
L3 – instead of "intermediate ice cover" write "varying ice cover"
Changed.

L8 – "… 19 April), **because is the time period when the ice retreated rapidly and there was more light availability**, despite the increasing gas transfer velocity."
We cannot be sure that there was increased light availability as the ice melted, because the mixed layer depth was also increasing. As mentioned above, we are no longer calculating GOP and NOP during this period so this section has been removed.

P21
L6 - "… especially **under thin** first-year ice…"
Changed.

L11 – this information on ice thickness becomes too late in the manuscript, move it to Pag. 5.
We have added it on page 5.

P22
L19 – title of section 3.4.2 has twice "of"
Done.

L26 – instead of "decreased" use "retreated"
Done.

P24
L3 – "… appears to decrease during …"
This paragraph, addressing explanations for the changes in GOP and NOP from 16-19 April has been eliminated as we no longer calculate GOP and NOP during this period.

L3 – as mentioned earlier, the changes in NOP appear to be contrasting when the transition of sea-ice melting period starts in 16 April, I would still strongly suggest to modify the GOP first period also to 16 April instead to until 19 April, so during that transition NOP decreases and GOP increases, and the only way that this can take place is if at the same time that GOP increases also respiration increases (net heterotrophy) as discussed below. Also the entrainment of deeper waters as the mixed layer deepens, contributes to changes in GOP during the transition period.
This paragraph, addressing the changes in GOP and NOP from 16-19 April has been eliminated.

However, in the section on GOP calculations, we now state:
*"This non-steady state equation for GOP incorporates temporal variability in both $O_2$ concentration and isotopic composition (see equation 5 in Prokopenko et al. (2011)). It does not account for time-variability in mixed layer depth (e.g. entrainment of deeper waters into the mixed layer), nor does it account for vertical diffusion of $O_2$ across the mixed layer. In order to*

*account for vertical processes in the O₂ mass balance, we would need to have triple oxygen isotope measurements below the mixed layer (Castro Morales et al., 2013; Munro et al., 2013;Wurgaft et al., 2013; Nicholson et al., 2014). Since we do not have these data, we cannot correct for vertical processes affecting the O₂ budget."*

L6-7 – what about the entrainment of low oxygenated waters as the mixed layer deepens?
Entrainment is now addressed, see our response to your previous comment.

L13-14 – the fact that there were no measurements below the mixed layer does not imply that vertical mixing is not an important process and should still be discussed.
This paragraph, addressing the changes in GOP and NOP from 16-19 April has been eliminated. As mentioned in response to your comment on L3, we now discuss mixing.

L16 – In Castro-Morales et al., 2013, the influence of entrainment and vertical mixing in mixed layer GOP and NOP are thoroughly discussed. This citation should be included here.
This paragraph, addressing the changes in GOP and NOP from 16--19 April has been eliminated.

L34 – what about nutrients sources from land (eutrophication), and also through Little Narrows during tidal exchange? How about a correlation between ice coverage and tides? Horizontal exchange in this region seems more relevant, warmer water entering WB influencing ice melt.

We do not have any nutrient data available during our time-series and therefore cannot address eutrophication impacts. We now added the following text in the new section "site description":
*"In general, excluding the narrow channels in direct contact with the ocean, the Bras d'Or Lakes system does not exhibit significant tidally-driven variability in temperature, salinity, and sea surface level (Krauel, 1975; Petrie, 1999; Petrie and Bugden, 2002; Dupont et al., 2003). A 21-day time-series in western Whycocomagh Bay showed no detectable diurnal or semidiurnal tides (Dupont et al., 2003)."*

P25
L23-26 – Residence time calculations might help supporting this statement
(t=h/k), please add this information considering the changes in k and h through the time series.

This sentence, regarding possible explanations for the decrease in NOP during/following ice melt, has been removed as Reviewer 2 requested we "substantially shorten the discussion of zooplankton and heterotrophy".

The residence time calculation you recommended was performed and included in the section on GOP results:
*"We believe that the O₂ measurements at Little Narrows from the beginning of the time-series through 16 April (the date when the fraction of open water in Whycocomagh Bay began to*

*increase rapidly) are generally representative of the under-ice rates due to the rapid transport of water through Little Narrows channel. The mean current speed at Little Narrows channel at 0.5 m depth was 3.4 km d⁻¹. The distance between the ice edge and the sampling intake ranged from ~0.4–1 km between the beginning of the time-series and 12 Apr. Assuming a gas transfer velocity for $O_2$ of 0.5 m d⁻¹ once the water mass encountered open water and a mixed layer depth between 0.8 to 2 m, the residence time of $O_2$ in the mixed layer was 1.6–4 d. Assuming a current speed of 3.4 km in open water, the water mass would have only been exposed to open water for between 0.1–0.3 d (20 % or less of the residence time of $O_2$). However, we recognize that the current speed near the ice edge may have been somewhat slower than the speed at the constriction at Little Narrows, therefore the transit time in open water may have been somewhat longer than 0.1–0.3 d."*

L26 – I agree in this potential explanation, considering that there might be low oxygenated water below the mixed layer in this region.
This sentence has been removed as we no longer calculate GOP during the period referenced here.

P26
L12 – " ~ 0) **on a daily basis**."
Done.

P27
L5 – "… but the salinity **in sea ice** is substantially lower … "
Done.

P28
L4 & 10 – add also to the list of references Castro-Morales et al., 2013
Done.

L29 – "… also from rivers, **or due to the entrainment of poor oxygenated waters**."
Done.  We reworded this sentence as:
*"The apparent decrease in NOP may be due to a number of factors, such as the onset of nutrient limitation, an increase in heterotrophic respiration of organic carbon, and/or the entrainment of waters with a lower $O_2$ concentration into the mixed layer (Castro Morales et al., 2013, Wurgaft et al, 2013, Nicolson et al, 2014)."*

Tables
- Include a small table (or embed this information in Table 1) where includes the following data: full experiment period with dates from start of the experiment (25 March) until the end (28 April), also when the injections took place, and not only the dates for determination of k. Include in the table also the ice coverage/fraction. It is hard now to follow all these changes between beginning of tracers experiments, beginning of the full experiment, evolution of ice coverage, etc.

We added in a new table with this information (copied below). We do not report the quantitative ice coverage/fraction in this table as it was not clear to us over what area this should be estimated. However, we do qualitatively describe changes in ice cover in this table.

**Table 1.** Key events during experiment

| Date (in 2013) | Event |
| --- | --- |
| 22 Mar | First CTD profile at Little Narrows |
| 25 Mar | Start of continuous water sampling with water pump and CT sensor at Little Narrows |
| 30–31 March | Tracer Injection 1 (under ice) |
| 7 April | ADCP installed at Little Narrows |
| 7–10 April | Tracer from Injection 1 sampled at Little Narrows |
| 16–19 April | Rapid decline in ice cover in Whycocomagh Bay |
| 19 April | Tracer Injection 2 (open water) |
| 20–23 April | Tracer from Injection 2 sampled at Little Narrows |
| 28 Apr | End of continuous water sampling and ADCP measurements at Little Narrows, and last CTD profile at Little Narrows |

Figures

Figure 1 – In panel b, add latitude and longitude values in axis, as in panel a. In panel c, add some labels, e.g., Greenland, Canada. The colors in this panel are too subtle and hard to see. On panel c (shown below) we have added labels for Canada, Greenland, United States and Cape Breton Island and changed the background colour to increase contrast as shown below. We have not added latitude and longitude to panel b as the figure was modified from an existing image that did not have latitude and longitude coordinates. As you mention, latitude and longitude are already given on Figure 1 panel a (as well as Figure 2) and we feel this is sufficient.

[Figure]

Figure 3 – Why the uppermost measurement did not start from the same depth ? I understand that maybe in the presence of ice this was not possible, but what about in open water?
We added to the figure caption *"The shallowest depth varies between casts due to challenges in getting stable CTD data in the upper 1 m."* We were not able to view the CTD data in real time and therefore did not know whether it had stabilized and had to correct for this later. On some casts we got stable measurements closer to the surface (typically if we left the CTD in the water for longer before lowering it).

Is there also a dashed vertical line in the last day of sampling (28 April)? Was a last CTD cast conducted that day?
Yes. We added to the figure caption (new text in italics) "The vertical grey dashed lines indicate the timing of CTD casts *(22, 26, 29, and 31 Mar, and 4, 6, 8, 12, 16, 20, 23, and 28 Apr)*."

At best, salinity "units" should be removed everywhere.
As discussed above, we have retained PSS when referring to salinity values. PSS is an acronym for Practical Salinity Scale and is not a unit. We feel that specifying PSS is necessary. There multiple salinity scales commonly used (ppt, g/kg, PSS, etc.), especially in estuarine science, and without this acronym it is not clear which scale was used.

Reference: Millero (1993) What is PSU? https://tos.org/oceanography/assets/docs/6-3_letter.pdf

Wasn't 25 of March the first day of sampling? Why there is data from even earlier than 24 March? Refer in the caption to Table 1 for specific dates of the vertical dashed lines.
Yes, we collected a CTD profile before the mass spectrometer was operational. We added to the figure caption (new text in italics) "The vertical grey dashed lines indicate the timing of CTD casts *(22, 26, 29, and 31 Mar, and 4, 6, 8, 12, 16, 20, 23, and 28 Apr)*." As mentioned above, we created a new Table 1 which lists the dates of the first and last CTD casts as well as other relevant dates.

Figure 4 – There is a typo in the legend for Injection 2, it should be 0.71 and not 0.071. In the line corresponding to Injection 1, the second to fourth measurements point (between 8th and 10th day) it looks actually an increasing tracers ratio. How can the authors explain this? In the end, the line for injection 1 was drawn by three points, initial, day 9 and day 11 (or end), would this still be valid?
Thank you for noticing this important error. We have corrected the figure as shown below.

The reason those points are increasing is likely a noise effect. We stated on page 10, line 23; "For Injection 1, the excess $SF_6$ and $^3$He concentrations were reduced by two orders of magnitude by the time the tracer reached Little Narrows (7–11 days after injection). The tracer ratio did not display a consistent decrease over the three days we sampled it at Little Narrows, likely in part due to the substantially lower gas transfer velocity."

[Figure]

Figure 6 – In the caption, it should be written at the end " … GOP term are most uncertain **due to melting of sea-ice cover**."

We incorporated your suggestion but reworded it as we no longer calculate GOP during this period:

On Figure 6: *"The shaded grey area is the period where we do not calculate NOP due to uncertainties related to the rapidly melting ice cover and the non-steady state term."*

On Figure 7: *"The shaded grey area is the period where we do not calculate NOP due to uncertainties related to the rapidly melting ice cover."*

Figure 7 – Would be good to also add in panels a and b the mixed layer depth using a secondary y-axis to contrast the time evolution of temperature and O2/Ar with MLD. In the caption it should be changed "yellow diamonds and pink triangles" to "green diamonds and blue squares" isn't?

We now include mixed layer depth on Figures 6 and 7. We removed panel from these figures (the cumulative NOP and GOP calculations) as we no longer calculate NOP and GOP through the whole time series and replaced this panel with the mixed layer depth.

We changed the caption to *"green diamonds and blue squares"*.

References
- P1, L16 - Here Loose and Schlosser, 2011 should be "2011b". Loose et al., 2011b should become 2011c.

We haven't changed this because we used the official Biogeosciences LaTeX template when preparing this manuscript. When we compiled the file, the references were automatically cited as *Loose and Schlosser (2011)*, *Loose et al. (2011a)* and *Loose et al. (2011b)*.

\- Add the doi's to all the references where it is missing
Done.

\- in Eveleth et al., 2016, change to full citation (not in press anymore), vol. 139, 89-102, 10.1016/j.dsr2.2016.07.016.
Done.

References listed in this review
Castro-Morales, K., N. Cassar, D. R. Shoosmith and J. Kaiser. Biological production in the Bellingshausen Sea from oxygen-to-argon ratios and oxygen tripe isotopes, 2013. Biogeosciences, Vol. 10, pag. 2273-2291.

Dupont, F., B. Petrie, J. Chaffey, 2003. Modeling the tides of the Bras d'Or Lakes, in: Canadian Technical Report of Hydrography and Ocean Sciences 230. Ocean Sciences Division, Maritimes Region, Department of Fisheries and Oceans, Bedford Institute of Oceanography, Dartmouth, Nova Scotia, Canada.
(http://www.bio-iob.gc.ca/science/research-recherche/ocean/webtide/documents/model_brasdor.pdf)

Lambert, T., 2002. Overview of the Ecology of the Bras d'Or Lakes with emphasis on the fish. Proc. N.S. Inst. Sci. Vol. 42, Part 1, pag. 65-99.

Luz B., Barkan E., Bender M.L., Thiemens M.H., Boering K.A. 1999. Triple-isotope composition of atmospheric oxygen as a tracer of biosphere productivity. Nature, 400, pag. 547–50.

Luz B., Barkan E. 2000. Assessment of oceanic productivity with the triple-isotope composition of dissolved oxygen. Science, 288, pag. 2028–31.

All of these papers except for the one by Lambert are now cited in the manuscript.

Thank you again for your thorough and constructive review of our manuscript. We have used your suggestions to improve our article.

---

## Author Response (AR1)

Dear Dr. Middelburgh:

We are pleased to submit our revised manuscript, "Changes in gross oxygen production, net oxygen production, and air-water gas exchange during seasonal ice melt in Whycocomagh Bay, a Canadian estuary in the Bras d'Or Lake system." Our author comments posted on Biogeosciences Discussions (https://www.biogeosciences-discuss.net/bg-2017-428/#discussion) contain a point-by-point response to the reviews and detail the changes made to the manuscript in response to each reviewer comment. A marked-up manuscript is included following this letter.

As recommended by Reviewer 1, we have changed the title of the manuscript, added a new section "site description" with new references to past work in the Bras d'Or Lake system, added a new table with the experimental timeline and modified Figure 1. As recommended by Reviewer 2, we provided an explanation of why the tracer ratio did not steadily decrease during injection 1, and added more discussion of the potential effect of the open water on the measured productivity at Little Narrows.

In response to comments from both reviewers relating to lack of subsurface measurements of the $O_2$ tracers, we decided to no longer calculate GOP or NOP during the ice melt period because of the uncertainties in the steady-state and non-steady state term at this time. We now more clearly state the uncertainties in our calculated GOP and NOP values due to the lack of subsurface measurements and explicitly recommend that researchers conducting similar studies in the future include depth-resolved measurements. We modified Figures 6 and 7 (GOP and NOP results) to include the mixed layer depth instead of the cumulative GOP and NOP.

Reviewer 2 recommended we revise our GOP calculations to use equation 1a from Haskell et al. (2017) in order to incorporate non-steady state $O_2$ concentration and mixed layer depth into the calculation. We explained in our author comment that the equation we used to calculate GOP already incorporates non-steady state $O_2$ concentration and that we have concluded that incorporating non-steady state mixed layer depth into the GOP calculation would require knowledge of the $O_2$ concentration and isotopic composition below the mixed layer, data which we did not collect. When we carefully derived an equation for GOP with a non-steady state mixed layer depth, we did not achieve the same equation as in Haskell et al. (2017). Therefore, the equation we used to calculate GOP remains unchanged.

We also made the text modifications and additions recommended by both reviewers. We would like to thank the reviewers again for their constructive and thorough reviews that have improved the manuscript.

Thank you very much for your patience as we worked to revise this manuscript, incorporating the recommendations of both reviewers.

Sincerely,
Cara Manning

[revised manuscript text omitted]

---

## Author Response (AR2)

Dear Dr. Middelburg:

Thank you very much for your careful review of our revised manuscript. We have made all of the changes you requested, except that we did not change "sea ice" to "sea-ice" in the abstract. There are eleven instances of "sea ice" in the manuscript and we would prefer to leave them all unhyphenated, however, if the Biogeosciences editorial style guide favours "sea-ice" then we give permission for all instances of "sea ice" in the manuscript to be changed to "sea-ice". We have provided a pdf version of the Supplement (as that was the only file format accepted through the website) but can provide the Supplement as a .tex file and the individual supplemental figure files upon request. Thank you.

Sincerely,
Cara Manning